# Dexamethasone-Induced Fatty Acid Oxidation and Autophagy/Mitophagy Are Essential for T-ALL Glucocorticoid Resistance

**DOI:** 10.3390/cancers15020445

**Published:** 2023-01-10

**Authors:** Miguel Olivas-Aguirre, Jesús Pérez-Chávez, Liliana Torres-López, Arturo Hernández-Cruz, Igor Pottosin, Oxana Dobrovinskaya

**Affiliations:** 1Laboratory of Immunology and Ionic Transport Regulation, Biomedical Research Centre, University of Colima, Av. 25 de Julio #965, Villas de San Sebastián, Colima 28045, Mexico; 2Medicine Faculty, University of Colima, Av. Universidad #333, Las Víboras, Colima 28040, Mexico; 3Department of Cognitive Neuroscience and National Laboratory of Channelopathies (LaNCa), Institute of Cellular Physiology, National Autonomous University of Mexico, Mexico 04510, Mexico

**Keywords:** acute lymphoblastic leukemia, glucocorticoids, glucocorticoid resistance, dexamethasone, autophagy, mitophagy, metabolic reprogramming, leukemic microenvironment

## Abstract

**Simple Summary:**

Reduced response to glucocorticoids (GCs, such as dexamethasone, Dex), first-line drugs employed during therapy induction, is considered a poor prognosis factor for T-cell acute lymphoblastic leukemia (T-ALL) patients. Here, the effects of Dex in patient-derived T-ALL cell lines were analyzed. Dex primarily targeted and rapidly accumulated in mitochondria, eventually causing a metabolic switch from glycolysis and glutaminolysis towards lipolysis and increased fatty acid oxidation (FAO), mitochondrial extra energization, and increased ROS production. Finally, mitochondrial damage/fission and autophagy/mitophagy were observed. Prevention of either FAO or autophagy greatly sensitized both T-ALL cells to Dex, which can be used to overcome GC resistance in T-ALL.

**Abstract:**

ALL is a highly aggressive subtype of leukemia that affects children and adults. Glucocorticoids (GCs) are a critical component of the chemotherapeutic strategy against T-ALL. Cases of resistance to GC therapy and recurrent disease require novel strategies to overcome them. The present study analyzed the effects of Dex, one of the main GCs used in ALL treatment, on two T-ALL cell lines: resistant Jurkat and unselected CCRF-CEM, representing a mixture of sensitive and resistant clones. In addition to nuclear targeting, we observed a massive accumulation of Dex in mitochondria. Dex-treated leukemic cells suffered metabolic reprogramming from glycolysis and glutaminolysis towards lipolysis and increased FAO, along with increased membrane polarization and ROS production. Dex provoked mitochondrial fragmentation and induced autophagy/mitophagy. Mitophagy preceded cell death in susceptible populations of CCRF-CEM cells while serving as a pro-survival mechanism in resistant Jurkat. Accordingly, preventing FAO or autophagy greatly increased the Dex cytotoxicity and overcame GC resistance. Dex acted synergistically with mitochondria-targeted drugs, curcumin, and cannabidiol. Collectively, our data suggest that GCs treatment should not be neglected even in apparently GC-resistant clinical cases. Co-administration of drugs targeting mitochondria, FAO, or autophagy can help to overcome GC resistance.

## 1. Introduction

Acute lymphoblastic leukemia (ALL) represents a group of aggressive hematological diseases and is the leading cancer type in children and the second leukemic type in adults. T-ALL arises from the malignant transformation of T cell precursors and possesses a highly proliferative and invasive phenotype. Although intensifying the therapeutic regimens has led to an important improvement in pediatric T-ALL outcomes, up to 60% of adult T-ALL can reach induction but not long-term remission [1]. Unfortunately, recovery has remained poor, with less than 25% survival rates for relapsed disease at all ages [2]. Conventional ALL therapy is based on a multidrug strategy, which consists of an induction phase where a combination of GCs (e.g., dexamethasone, Dex; prednisone), antimitotic drugs (e.g., vincristine) and antimetabolites is applied, followed by an intensification/consolidation phase, which includes asparaginase, methotrexate, cyclophosphamide, cytarabine, and the use of intrathecal chemotherapy and cranial radiation [3,4,5]. Despite the wide variety of drugs employed, GCs belong to the first-line treatment, and early response to GCs represents an initial prognosis factor. Several clinical trials have supported the preferential use of Dex in treating T-ALL over prednisone [2]. GC resistance is considered an unfavorable prognostic factor, and limited GC response in ALL patients has been associated with an elevated risk of minimal residual disease and poor survival [6,7].

The effects of GCs are traditionally attributed to the binding and activation of cytosolic receptors (glucocorticoid receptors, GRs), their translocation to the nucleus, and, through binding to the GC response elements (GRE), the transactivation or transrepression of multiple genes. In ALL therapy, GC sensitivity implies the capacity of GCs to induce cell death in leukemic cells. The most-studied and discussed mechanism of GC lymphotoxicity is the transactivation of genes encoding pro-apoptotic proteins (e.g., *BCL2L11* encoding BIM) and the activation of apoptotic pathways. Accordingly, GC resistance in T-ALL is often attributed to aberrant GR expression (decreased or null-expression, presence of non-functional GR isoforms), hypermethylation of genes coding for effector proteins (e.g., BIM), and overexpression of multidrug resistance proteins, MDR [8]. However, expression of aberrant GRs is usually found in cell lines derived from relapsed patients and additionally selected for resistance to GC during long-term exposure to high GC concentrations, but is rarely detected in primary samples of GC-resistant T-ALL [9,10].

GCs generate a massive transcriptional response that affects up to 5558 genes and more than 1151 transcripts [11]. Transcriptional profiling of GC-resistant T-ALL revealed additional factors, independent of GR expression, that can contribute to resistance: activation of glycolysis and glutaminolysis, increased oxidative phosphorylation (OXPHOS), and activation of PI3K/AKT/mTOR and MYC signaling pathways [9]. Considering the re-programming of cancer cells towards glycolytic metabolism, GC-dependent downregulation of metabolic proteins such as glucose transporters and glycolytic enzymes, among others, is essential [8].

Mitochondria have been proposed as emerging GC targets since they possess standard GRE in their genome [12,13]. Some of the rapid effects of GCs have been attributed to the mitochondrial site of action [14]. GRs in mitochondria of different cell types, GRE in mitochondrial DNA sequences, and translocation of GC–GR complexes to mitochondria have been reported [15,16,17]. Different isoforms likely trigger different signaling pathways to interact during the cellular response to GC. Additionally, a non-genomic mechanism has been revealed, which consists of direct interaction between GC–GR complexes and pro-apoptotic proteins in the mitochondrial membrane [17].

In this study, we have addressed the Dex uptake, intracellular localization, and retention by T-ALL cells, its effect on mitochondrial morphology and function, the metabolic switch, and autophagy/mitophagy induction to achieve a better understanding of how these changes support the GC-resistant vs. GC-sensitive phenotype and to find a way to revert GC resistance in T-ALL.

## 2. Materials and Methods

### 2.1. Reagents

The following reagents were purchased from Sigma-Aldrich (San Luis, MO, USA): carbonyl cyanide m-chlorophenyl hydrazone, CCCP (Cat. #C2759), chloroquine diphosphate salt, CQ (Cat. #C6628), curcumin (Cat. #C7727), dexamethasone, Dex (Cat. #D9184), mifepristone, Ru486 (Cat. #M8046), N-acetyl-L-cysteine, NAC (Cat. #A7250), probenecid (Cat. #57669), and sulfinpyrazone (Cat. #57965). Etomoxir (Cat. #11969) and cannabidiol, CBD (Cat. #90081) were acquired from Cayman Chemicals (Ann Arbor, Michigan, USA). For stock solutions, drugs were dissolved in methanol, ethanol, or DMSO (Cat. #154903, #E7148, and #D4540, respectively). At the maximum estimated drug concentrations, the content of solvents did not exceed 1% of the total sample volume and did not affect cell viability.

### 2.2. Cell lines and Culture Conditions

All cell lines used in this study were from the American Type Culture Collection (ATCC; Manassas, VA, USA). All growth media and supplements were purchased from Gibco, Thermo Fisher Scientific, Fair Point, NY, USA. Human T-ALL cell lines, Jurkat (ATCC^®^TIB™, Clone E6-1, male, 14 years), MOLT-3 (ATCC^®^CRL-1552™, male, 19 years), and CCFR-CEM (ATCC^®^CCL-119™, female, 4 years), were grown in suspension in advanced RPMI 1640 medium, supplemented with 5% (*v*/*v*) of heat-inactivated fetal bovine serum (FBS), 100 U/mL of penicillin, 100 µg/mL streptomycin, and 1% of GlutaMAX™. Human breast cancer cell lines MDA-MB-231 (ATCC^®^HTB-26™, female, 51 years) and MCF7 (ATCC^®^HTB-22™, female, 69 years), as well as human embryonic kidney cells HEK-293 cells (ATCC^®^CRL-1573™), were maintained in DMEM medium supplemented with 10% (*v*/*v*) of heat-inactivated FBS, 100 U/mL of penicillin, 100 µg/mL streptomycin, and 1% of GlutaMAX™. Medium alpha-MEM supplemented with 5% of fetal bovine serum (FBS), 2 mM of GlutaMAX, 10 mM of HEPES, 100 U/mL penicillin, and 100 μg/mL of streptomycin was used for murine mesenchymal stromal cells (MSC) from bone marrow (BM), OP9 (ATCC^®^ CRL-2749™). OP9 cells stably expressing green fluorescent protein (OP9-GFP) were a kind gift from Dr. Rosana Pelayo (CIBIOR-IMSS, Puebla, Mexico). OP9-GFP cells were cultured as wild-type OP9 cells. All cell lines were used for experimentation during the first 20 culture passages and maintained in a humidified incubator at 37 °C with 5% CO_2_.

### 2.3. Resazurin-Based Metabolic (Cytotoxicity) Assay

Cells (1.5 × 10^6^ cells/mL) were cultured for the determined periods (0–96 h), collected, centrifuged, resuspended in a fresh media, and placed in a 96-well plate (180 μL/sample). Subsequently, 20 μL of resazurin reagent (Tox-8, Sigma-Aldrich) was added to each well, and the mixture was further incubated for 2 h. During incubation, non-fluorescent resazurin is metabolized by intracellular enzymes into resorufin, a highly fluorescent compound (Ex/Em max = 560/590 nm). Resorufin production by viable cells was estimated using a GloMax plate reader (Promega, Madison, WI, USA). The samples were excited at 525 nm, and the emitted fluorescence signal was recorded at 580–640 nm. The remaining fluorescence of the medium was subtracted. Collected data were normalized to control and graphed as an average of at least three independent experiments.

### 2.4. Assessment of Cell Viability by Cell Count (Trypan Blue Exclusion Test)

Control or treated cells (1.5 × 10^6^/mL) were cultured in 48-well plates. After the incubation period (0–96 h), each sample was carefully resuspended, and a 10 μL aliquot was mixed with 10 μL of trypan blue dye solution (Sigma-Aldrich, Cat. #15250061). Then, 10 μL of the mixture was placed in a Neubauer chamber to count live (unstained) and dead (blue-stained) cells. Cell counts from at least three independent experiments were averaged and plotted. A Nikon D750 camera was adapted to a Nikon light microscope for cell counting to acquire representative images of cell cultures and treatments.

### 2.5. Cell Death Analysis by Flow Cytometry

The Dead Cell Apoptosis Kit (Thermo Fisher Scientific, Cat. #V13241) was employed to evaluate the cell death induced by Dex as recommended by the manufacturer, with some modifications. The kit is based on the detection of apoptotic cells by using a recombinant annexin V conjugated with a fluorescent dye, Alexa 488 (A488), and a nucleophilic marker to establish necrosis (propidium iodide, PI). Cells maintained in a complete growth medium (1.5 × 10^6^ cells/mL) were treated with Dex (0–10 μM) for 24 h. Then, cells were resuspended, counted, and centrifuged (5 min, 400× *g*). The supernatant was discarded, and the pellet containing cells was resuspended in the working solution (annexin-binding buffer) containing 2 μL of Annexin-V and 1 μL of PI (200 μg/mL), and incubated for 20 min. After the incubation period, cells were examined using flow cytometry (FACS Canto II) to determine the inclusion of cell death indicators. A 488 nm laser was used for the excitation of both fluorochromes, A488 (Ex/Em max = 488/510 nm) and PI (Ex/Em max = 535/617 nm). Emitted fluorescence was measured using a 502LP mirror and 530/30 filter for A488, and a 556LP mirror and 585/42 filter for PI. The raw data containing 10,000 events in a single-cell gate (doublets and debris were gated out) were further analyzed using FlowJo software (v 7.0). The percentage of dead cells (A488+/PI−, A488−/PI+, and A488+/PI+) in the whole cell population was estimated. Data from at least four independent experiments were averaged and plotted.

### 2.6. Proliferation Assay by Flow Cytometry

Carboxy-fluorescein succinimidyl ester (CFSE; Thermo Fisher Scientific, Cat. #C34554) was employed to determine the proliferation rate of T-ALL cells. A previously described protocol [18] with modifications for suspension cell lines [19] was used. Cells were collected, centrifuged, and resuspended in PBS (1 × 10^6^/mL). For staining, 0.5 μM of CFSE was added, and samples were incubated for 30 min at room temperature and agitated every 5 min. After incubation, non-incorporated CFSE was removed by washing with PBS. Stained cells were resuspended in the fresh growth medium and seeded in a 24-well plate (1.5 × 10^5^ cells/mL). Dex was added at this point, and cells were incubated for up to 96 h. Control and treated cells were collected every 24 h, washed with PBS, and CFSE fluorescence was evaluated using flow cytometry (FACS Canto II, BD Biosciences, Franklin Lakes, NJ, USA) and analyzed and plotted using FlowJo software. A 488 nm laser was used for excitation, and emitted fluorescence was recorded with a 530/30 nm filter. 10,000 events corresponding to the single-cell gate were analyzed for each sample. The mean CFSE fluorescence intensity (MFI) of Dex-treated cells was normalized to the MFI value in the corresponding control. Data are the average of at least four independent experiments.

### 2.7. Dex Uptake Assay

Fluorescein-conjugated Dex analog (DexFluo, Ex/Em max. = 490/525 nm; Thermo Fisher Scientific, Cat. #D1383) was used to monitor Dex incorporation into T-ALL cells and its intracellular localization. Cells (1.5 × 10^6^/^mL^) were incubated in the presence of DexFluo (0–10 µM) for 24–72 h. After the incubation period, cells were harvested and washed twice with fresh PBS to remove the excess of extracellular Dex. Cells were further assayed using flow cytometry (FACS Canto II cytometer, BD Biosciences, Franklin Lakes, NJ, USA) or confocal microscopy (LSM 700, Carl Zeiss, Jena, Germany). For cytometry data acquisition, samples were excited with a 488 nm laser, and emitted fluorescence was collected using a 530/30 nm filter. Raw data were gated to eliminate cell debris and doublet events. A total of 10,000 events in the gates of single cells were acquired from each sample, and each population’s mean fluorescence intensity was normalized to control cells. The average of at least three independent experiments was plotted as the Dex uptake rate. For imaging, cells were placed into coverslip-bottomed chambers and 40×, 63×, and 100× oil-immersion objectives were used. A 488 nm laser was used for excitation. Images were generated with Zen lite 3.0 software (Carl Zeiss, Jena, Germany). Raw images were further processed using ImageJ 1.53t software (NIH).

### 2.8. Glucose Uptake Assay

The fluorescent analog of glucose 2-(*N*-(7-nitrobenz-2-oxa-1,3-diazol-4-il) amino)-2-desoxyglucose (2-NBDG, Ex/Em max. = 465/540 nm; Thermo Fisher Scientific, Cat. #N13195) was employed to estimate the effect of Dex on glucose retention by T-ALL cells. Cells (1.5 × 10^6^/mL) were seeded in a 48-well plate in a glucose-free RPMI medium in the presence of different 2-NBDG concentrations (0–200 μM), and incubated either for 30 or 60 min. After incubation, the cells were collected and washed twice with fresh PBS to eliminate the non-incorporated 2-NBDG. The incorporation of 2-NBDG was further validated using flow cytometry by collecting 10,000 events per sample. Significant changes in 2-NBDG incorporation were observed from 5 μM and remain indistinguishable up to 20 μM. Maximum 2-NBDG retention was observed with 200 μM. Representative histograms of the fluorescence of each sample are presented in Appendix A. To validate that the 2-NBDG incorporation mechanism is linked to traditional glucose transport through glucose transporters (GLUTs), a competition assay was performed by adding glucose (non-fluorescent) to cells exposed to 2-NBDG. Based on 2-NBDG titration, 5 μM of 2-NBDG and 1 h incubation was employed for this and subsequent experiments. In the presence of glucose (10–20 mM), 2-NBDG retention was limited (Appendix A).

For the experiments, cells (1.5 × 10^6^/mL) were seeded in a 48-well plate, exposed to various concentrations of Dex, and further incubated for 24–48 h in a complete RPMI medium. After the incubation period, cells were harvested, counted, centrifuged, resuspended, and cultivated in a glucose-free RPMI medium for 2 h to promote cell starvation and stimulate subsequent glucose uptake. At the end of the incubation period, 5 μM 2-NBDG (concentration based on flow cytometry titration curve) was added, and cells were allowed to take it up for 1 h. After incubation, the cells were harvested, washed twice with PBS to remove non-incorporated 2-NBDG, and resuspended in fresh PBS. Samples were then placed in a 96-well plate, and 2-NBDG fluorescence was assessed using a GloMax Discover plate reader (Promega, Madison, WI, USA) by excitation at 475 nm and collecting emitted fluorescence at 500–550 nm. Raw data were normalized to control, averaged, and graphed as a percentage of glucose uptake by Dex-treated cells compared to untreated cells. The presented data are the average of at least three independent experiments.

### 2.9. Glutamate Production Assay

To determine the rate of glutaminolysis, the total glutamate production was assessed using the Glutamate-Glo™ test (Cat. #J8021, Promega, Madison, WI, USA), with some modifications. Briefly, as recommended by the manufacturer, cells were cultured in a complete growth medium for 0–72 h in the presence or absence of Dex (1 μM). After incubation, cells (5 × 10^5^/mL) were harvested, centrifuged, and resuspended in a fresh medium, free of L-glutamine, to avoid a non-specific glutamate production. The glutamate detection reagent (50 µL), containing glutamate dehydrogenase, NAD^+^, reductase, pro-luciferin reductase substrate, and Ultra-Glo™ recombinant luciferase, was added to cells in a 1:1 *v/v* ratio and incubated for 30 min. Luciferin is produced due to an enzymatic reaction, and its luminescent signal is proportional to the amount of glutamate in the sample. Luminescence of luciferin was recorded over a 1 s period using a plate-reading luminometer (GloMax, Promega, Madison, WI, USA). Raw data from at least three samples were averaged. To validate the method’s sensitivity, cell-free RPMI samples were evaluated with added glutamate (25 μM).

### 2.10. Glycerol Production Assay

Total glycerol (intracellular and extracellular) was estimated using the Glycerol-Glo^TM^ assay to determine the glycerol production rate, indicative of lipolysis (Cat. #TM599, Promega, Madison, WI, USA). Briefly, as recommended by the manufacturer, cells (1 × 10^5^/mL) were cultured in the complete medium for 0–72 h in the absence or presence of Dex (1 μM). After the incubation period, 20 × 10^4^ cells were collected in 25 μL of RPMI medium, 25 μL of the glycerol lysis solution was added, and samples were incubated for 30 min. After incubation, 50 μL of the glycerol detection reagent was added (containing glycerol-3-p-dehydrogenase, NAD+, reductase, pro-luciferin reductase substrate, kinetic enhancer, and Ultra-Glo™ recombinant luciferase), and samples were incubated for 60 min at 37 °C. After the incubation period, luminescence was recorded using a plate reader (GloMax, Promega, Madison, WI, USA). Raw data from at last four independent experiments were averaged and graphed. Cell-free RPMI samples were used as a negative control to validate the method sensitivity, and glycerol (80 μM) was used as a positive control.

### 2.11. Lactate Production Assay

To determine the rate of glycolysis, total L-lactate production was assessed using the Lactate-Glo™ test (Cat. # J5021, Promega, Madison, WI, USA), as recommended by the manufacturer, with some modifications. Briefly, cells were cultured in the complete growth medium for 0–72 h, in the absence or presence of Dex (1 μM). After incubation, cells (5 × 10^5^/^mL^) were harvested, centrifuged, and resuspended in the fresh medium. The L-lactate detection reagent (50 µL), containing lactate dehydrogenase, NAD^+^, reductase, pro-luciferin reductase substrate, and Ultra-Glo™ recombinant luciferase, was added to cells in a 1:1 *v/v* ratio and incubated for 30 min. Luciferin is produced due to an enzymatic reaction, and its luminescent signal is proportional to the amount of L-lactate in the sample. Luminescence of luciferin was recorded over 1 s period using a plate-reading luminometer (GloMax, Promega, Madison, WI, USA). Raw data from at least three samples were averaged.

### 2.12. Evaluation of Mitochondrial Status by Mitochondria Staining

The fluorescent mitochondrial indicators used were MitoTracker^TM^ Red (MtRed, Ex/Em max = 581/644 nm; Thermo Fisher Scientific, Cat. # M7512) or tetramethyl rhodamine ethyl ester (TMRE, Ex/Em max = 555/582 nm; Thermo Fisher Scientific, Cat. # T669), dependent on the mitochondrial potential (ΔΨm), or ΔΨm-independent MitoTracker^TM^ Green (MtGreen, Ex/Em max = 490/518 nm; Thermo Fisher Scientific, Cat. # M7514). Staining was performed as described previously [20]. For imaging, cells were placed in home-made coverslip-bottomed chambers and analyzed using confocal microscopy (LSM 700, Carl Zeiss, Jena, Germany). Alternatively, fluorescence intensity was measured using a GloMax Discover plate reader (Promega, Madison, WI, USA), with the following configurations: (a) 620 nm excitation filter and 580–640 nm emission filter for MtRed; (b) 520 nm excitation filter and 580–640 nm emission filter for TMRE; and (c) 475 nm excitation filter and 500–550 nm emission filter for MtGreen. Raw data from at least three experiments were averaged, normalized against the control, and plotted.

### 2.13. Estimation of Mitochondrial Ca^2+^ by CEPIA3mt

Jurkat cells were transfected with a Ca^2+^-sensitive, mitochondria-targeted indicator CEPIA3mt (Ex/Em max = 488/510 nm) to monitor mitochondrial Ca^2+^, as described previously [20]. Transfected cells were treated with Dex (1 µM, 24 h) and analyzed using confocal microscopy (LSM 700, Carl Zeiss, Jena, Germany). Raw images were processed in ImageJ 1.53t software (NIH). For image analysis, regions of interest were determined using the software’s ROI manager by selecting the area of each cell on the brightfield image. The fluorescence of 100 cells for each condition was collected, quantified, averaged, and plotted.

### 2.14. Simultaneous Monitoring of Changes in Cytosolic and Mitochondrial Ca^2+^

The fluorescent Ca^2+^ indicators Fura-2-AM with Ex/Em max = 340/510 nm for Ca^2+^-bound form (Thermo Fisher Scientific, Cat. # F1201) and Rhod-2-AM (Ex/Em max = 552/581 nm; Thermo Fisher Scientific, Cat. #R1244) were used. Leukemic cells were loaded simultaneously with both indicators, as described previously [20]. Loading conditions for the predominant mitochondrial localization of Rhod-2 were monitored microscopically using the mitochondrial tracker MtGreen, as previously described [20]. Samples were placed in a quartz cuvette and evaluated using an F7000 spectrofluorometer (Hitachi High-Technologies, Tokyo, Japan). Changes in fluorescence corresponding to each indicator were alternatively recorded (2.5 s cycle) using excitation at 340 nm and collecting the fluorescence at 510 nm for Fura-2, and excitation at 552 nm and collecting the fluorescence at 581 nm for Rhod-2.

### 2.15. Estimation of ROS Production

To evaluate ROS production, the non-fluorescent dye 2′, 7′-Dichlorodihydrofluorescein diacetate (DCFH-DA; Sigma-Aldrich, Cat. #D6883) was used. DCFH-DA is changed to highly fluorescent DCF (Ex/Em max ≅ 492–495/517–527 nm) when oxidized by intracellular ROS. Control and Dex-treated cells (1 μM, 24 h) were collected, washed with PBS and stained with DCFH-DA (20 μM, 30 min) to determine ROS production. Then, cells were rewashed to remove the non-incorporated dye and resuspended in PBS. Samples were placed in a 96-well plate, and DCF fluorescence was evaluated using a plate reader (GloMax, Promega, Madison, WI, USA) with a 475 nm excitation filter and a 500–550 nm emission filter. Data from at least four independent experiments were averaged and normalized to control. Representative images were obtained through confocal microscopy (LSM 700, Carl Zeiss, Jena, Germany) using a 488 nm laser for DCF fluorescence excitation and 40×, 63×, and 100× oil-immersion objectives were used. Images were generated with Zen lite 3.0 software (Carl Zeiss, Jena, Germany). Raw images were further processed using ImageJ 1.53t software (NIH).

### 2.16. Estimation of Autophagic Flux

An acidotropic fluorescent dye monodansylcadaverine (MDC, Ex/Em max = 365/525 nm; Sigma-Aldrich, Cat. #30432) was employed to estimate the effects of Dex on the autophagic flux. MDC targets acidic compartments such as endosomes, lysosomes, and late-stage autophagolysosomes. MDC retention is upregulated when the formation of high-cargo autophagolysosomes is increased [21] and is commonly employed as an indirect strategy to evaluate autophagic flux. Cells treated with Dex (1 μM) for 0–72 h were collected, washed, resuspended in PBS, and incubated with MDC (60 μM, 30 min). Stained cells were rewashed to remove the excessive (non-incorporated) dye and placed in a 96-well plate. MDC fluorescence was evaluated using a plate reader (GloMax, Promega, Madison, WI, USA) with a 365 nm excitation filter and a 500–550 nm emission filter. Representative images were obtained through confocal microscopy (LSM 700, Carl Zeiss, Jena, Germany) using a 405 nm laser for MDC fluorescence excitation. Oil-immersion objectives 40×, 63×, and 100× were used. Images were generated with Zen lite 3.0 software (Carl Zeiss, Jena, Germany). Raw images were further processed using ImageJ 1.53t software (NIH).

Another acidotropic dye, Lysotracker Yellow HCK123 (Thermo Fisher Scientific, Cat. #L12491), was employed to confirm the obtained data, with a working concentration 100 nM and acquisition settings as follows: Ex max = 465 nm and Em max = 535 nm, corresponding to Lysotracker characteristics. Data from at least four independent experiments were averaged and normalized to the control group.

For comparison, the autophagic flux caused by Dex (1 μM; 24 h) was analyzed in MDA-MD-231 and OP9, large adherent cells with multiple mitochondria, stained with Lysotracker Yellow HCK123 or MDC. Raw data obtained through confocal microscopy (LSM 700, Zeiss, Jena, Germany) were further processed using ImageJ 1.53t software (NIH) to determine changes in autophagic flux by measuring either the number of particles per cell (dye puncta) or particle size (dye area). Such values were obtained by thresholding the raw images to remove the background signal and applying the “watershed” tool to separate individual objects. Finally, the “particle analysis” tool was employed to determine the number of particles, with a size between 0.5–2 μm^2^. The data obtained from at least five independent samples were averaged and further graphed. Cells were co-stained with Mt Red (200 nM, 30 min) and analyzed using confocal microscopy to explore the relation between the lysosomal content and the mitochondrial distribution.

### 2.17. pHAGE-mt-mKeima Transformation

DH5α competent bacteria (18258012, Thermo Fisher Scientific) were transformed by heat shock. A pHAGE-mt-mKeima construct [22] was added and incubated for 14 h at 37 °C in LB agar (Thermo Fisher Scientific, Cat. #22700025), supplemented with 100 μg/μL of ampicillin (Thermo Fisher Scientific, Cat. #11593027). Colonies were selected and transferred to a supplemented medium for further bacterial growth for 14 h. Plasmidic DNA was purified using a NucleoBond XtraMidi (740410.10, Macherey-Nagel) kit, and DNA purity and concentration were evaluated spectrophotometrically through absorption at 260/280 nm.

For transfection, 10^5^ T-ALL cells were starved in an Optimem reduced medium for 12 h, then exposed to liposomes composed of Lipofectamine 3000 (Thermo Fisher Scientific, Cat. # L3000015) and plasmidic DNA (pHAGE-mt-mKeima, 1 μg), and centrifuged for 30 min (400× *g*) to promote interaction between cells and the liposomes. Transfected cells were incubated in a serum-free growth medium overnight (37 °C, 5% CO_2_), and FBS (10%) was added the next day. Experiments were performed 24 h after transfection with untreated and Dex-treated (1 µM, 24 h) cells.

### 2.18. Mitophagy Analysis in Leukemic Cells Transfected with pHAGE-mt-mKeima Construct

Data analysis was based on the fact that the maximum fluorescence excitation of the mtKeima protein occurs at 586 nm at acidic pH and 440 nm at neutral pH, while the emission spectrum is independent of pH, with a maximum at 620 nm [23]. Accordingly, images were acquired using a confocal microscope (LSM 700, Carl Zeiss, Jena, Germany) equipped with 405 and 555 nm lasers and 40×/63×/100× oil-immersion objectives. Images were generated with Zen lite 3.0 software (Carl Zeiss, Jena, Germany). Raw images were further processed using ImageJ 1.53t software (NIH). For quantitative analysis, transfected cells were analyzed through spectrofluorometry (F7000 spectrophotometer, Hitachi High-Technologies, Tokyo, Japan). The samples were excited in the 300–700 nm range, and the fluorescence intensity values at 620 nm were estimated. The ratio, obtained at 620 nm for 586 nm and 440 nm excitation (586/440), reflected the mitophagy progression. Data were normalized to control (untreated) samples and averaged, and the data from at least four independent experiments were graphed using a GraphPad prism.

### 2.19. Transmission Electron Microscopy

Control and Dex-treated (1 µM, 24 h) Jurkat cells were washed (400× *g*), fixed with 2.5% glutaraldehyde, and post-fixed in 1% OsO_4_ and 0.8% K_4_Fe (CN)_6_·3H_2_O, and 5 mM Ca^2+^. Then, cells were dehydrated in acetone and embedded in Epon. Ultrathin sections were stained with uranyl acetate and lead citrate. Samples were examined using a JEOL JEM 12 000 EII transmission electron microscope at the Unidad de Imagenología, Instituto de Fisiología Celular (IFC), UNAM, Mexico City.

### 2.20. Bidimensional Co-Culture of MSCs and T-ALL Cells

A two-dimensional (2D) co-culture (2D-CC) system was developed as follows: MSC OP9-GFP cells (2.5 × 10^4^) were seeded on a custom-made experimental chamber in an alpha-MEM medium and cultured for 48 h to create a homogenous monolayer. Leukemic CCRF-CEM cells were collected, counted, and stained with Cell Tracker™ Deep Red (Ex/Em max = 647/668; Thermo Fisher Scientific, Cat. #C34565; 25 nM, 30 min). Stained leukemic cells were washed twice, added in a 1:1 ratio to MSCs, and treated with Dex (0–10 μM, 24 h). At the end of the incubation period, the supernatant was carefully collected, and the culture was gently washed twice with PBS (500 μL). The entire supernatant, which contained leukemic cells in loose contact with MSCs, was centrifuged, and the pellet was resuspended in 1 mL PBS for cell counting. The remaining monolayer, containing MSCs and tightly interacting leukemic cells, was analyzed using a confocal microscope equipped with 40×, 63×, and 100× oil-immersion objectives (LSM 700, Carl Zeiss, Jena, Germany). The GFP was excited with a 488 nm laser and Deep Red with a 639 nm one. The live-to-dead cells ratio by field (5 random micrographs for each condition) was estimated in 6 independent experiments. Only cells positive for Deep Red were considered to exclude non-adherent OP9 cells. Image analysis was performed with Image J (NIH).

### 2.21. Tridimensional Co-Culture of MSCs and T-ALL Cells

For the development of three-dimensional (3D) co-culture (3D-CC), a previously published protocol was used [24], with some modifications. OP9-GFP cells (2.5 × 10^4^) were seeded in a 96 round-bottom well plate coated with agarose (1%). Agarose limits adhesion and promotes the self-association of OP9 cells and spheroid formation. The spheroids were cultured in an alpha-MEM medium for 48 h to achieve shape uniformity. After that, CCRF-CEM cells stained with Cell Tracker™ Deep Red (25 nM, 30 min) were added to the spheroids and treated with Dex (0–10 μM; 24 h). After incubation, the plate with the intact 3D-CC was placed in an inverted light Nikon microscope, and images were acquired with a Nikon D750 DSLR camera (10× objective). Raw images were digitalized and analyzed using Image J. For confocal microscopy (LSM 700, Carl Zeiss, Jena, Germany), the 3D-CC was gently suspended, collected using custom-made pipette tips, and mounted in a VECTASHIELD^®^ antifade mounting medium (Vector Laboratories; H-1000-10). Raw data were acquired using 40×, 63×, and 100× oil-immersion objectives. GFP fluorescence was excited with a 488 nm laser, and Deep Red fluorescence with a 639 nm laser. Image J was employed for image processing and analysis.

### 2.22. Transwell Migration Assay

T-ALL cells and MSCs spheroids were seeded in 12-well plates (Transwell system, Corning Inc., Cat. #3422) and treated for 24 h with Dex (1 μM). T-ALL cells (2.5 × 10^4^) were placed in 200 µL of a serum-free alpha-MEM medium in the upper chamber of Transwell inserts (8 μm pore size). OP9 spheroids or recombinant human CXCL12 (100 ng/mL; Sigma-Aldrich) in the lower chamber, filled with complete (10% FBS) medium, were used to attract T-ALL. Cells were allowed to migrate for 24 h. After incubation, the insert was removed, and the cells in the lower chamber were moderately resuspended and counted. Raw data from 4 independent experiments were pooled, and migration was expressed as a percentage of T-ALL cells that migrated to the lower chamber of the total cell number.

## 3. Results

### 3.1. Two T-ALL Cell Lines Display Contrasting Sensitivity to Dex

#### 3.1.1. Dex Affects the Metabolism and Viability of CCRF-CEM but Not of Jurkat Cells

Cell lines derived from leukemia patients are widely used as suitable models for studying the biology of leukemic cells and their resistance to treatment. In the present work, we used Jurkat and CCRF-CEM cells derived from relapsed patients. Jurkat was reported earlier as GC-resistant, whereas unselected parental CCRF-CEM represents the heterogeneous population of GC-resistant and GC-sensitive clones, further referred to as “partly sensitive” [25,26]. Firstly, we validated their sensitivity to Dex using a resazurin-based metabolic assay and live cell count (Figure 1a–f). As expected, at 24 h after Dex administration, the metabolic level of CCRF-CEM cells decreased (Figure 1a, red trace). In contrast, the metabolism in the GC-resistant Jurkat cell population was not affected by Dex, even at high concentrations (Figure 1b, red trace). Remarkably, Dex did not affect the total number of cells in the populations of both tested cell lines during the first 24 h (Figure 1a,b; black traces). Since the maximum inhibitory effect in CCRF-CEM cells was produced by 1 μM Dex, this concentration was subsequently used to monitor longer-term effects in both cell lines. Dex-induced metabolic suppression was evident over time in CCRF-CEM cells (up to 96 h of observation), while Jurkat’s metabolism remained at the control level (Figure 1c,d). Accordingly, the live cell count decreased in CCRF-CEM compared to control, with a significant reduction after 72 h of Dex administration, but not in the Jurkat cells (Figure 1e,f). These observations were also confirmed by using an apoptosis/necrosis cell death assay, where DEX 0–10 μM administration caused dose-dependent cell death in CCRF-CEM, but not in Jurkat cells after 72 h of incubation (Figure 1g–i). Thus, in partly sensitive CCRF-CEM cells, Dex initially (first 24 h) significantly reduced metabolism. Strikingly, at longer times (72 and 96 h), when about half of the CCRF-CEM cells died (sensitive population) and the total number of living cells at 96 h was almost one order of magnitude less than the control, the total metabolic activity compared to that of the control cells was decreased by only one third (Figure 1c,e,h).

#### 3.1.2. GR Mediates the Effects of Dex on CCRF-CEM Cells

To determine whether the effects of Dex, observed in the sensitive population of CCRF-CEM cells, are mediated by GR, we employed a high-affinity GR antagonist, Ru486 (1 μM). Pre-incubation with Ru486 (20 min) prevented the effect of Dex (1 μM) as measured by both metabolic assay and cell count in CCRF-CEM cells, indicating that these damaging processes are triggered via GR (Figure 1j–m). As expected, no changes were observed in Jurkat cells treated with either Dex, Ru486, or their combination (Figure 1k,m).

#### 3.1.3. Dex Inhibits the Proliferation of CCRF-CEM Cells

Since Dex diminished CCRF-CEM living cell count as compared to the control, we aimed to study the effect of Dex on cell proliferation. Cell division was monitored using flow cytometry, measuring the fluorescence of the intracellular non-toxic dye CFSE. In this assay, cell division resulted in the CFSE distribution between daughter cells and decreased cell fluorescence intensity. Median fluorescence intensity (MFI), measured in the cell population, progressively shifted to a lower intensity as the cells divided (Figure 1n). Dex significantly decreased T-ALL CCRF-CEM cell proliferation at 96 h, whereas Jurkat proliferation remained unaffected at all times (Figure 1n).

### 3.2. T-ALL Cells Can Efficiently Take Up and Retain Dex

#### 3.2.1. Jurkat and CCRF-CEM Cells Retain Dex Differently

GC resistance may be due to the inability of cells to take up and retain Dex. To track Dex incorporation, its subcellular localization, and retention over time, a fluorescent Dex analog DexFluo was used.

Fluorescent microscopy analysis showed that one hour after incubation with the DexFluo (5 μM) administration, there was significant uptake of DexFluo, especially by CCRF-CEM cells as compared to Jurkat (Figure 2a). However, after 24 h, the DexFluo fluorescence level was approximately the same in both cell lines, as estimated using flow cytometry (Figure 2b, left panel). Further flow cytometry monitoring of the DexFluo fluorescence in the treated cells revealed that, in the period from 24 to 96 h, its level in CCRF-CEM cells was relatively constant. In contrast, it gradually decreased in Jurkat cells (Figure 2b). Thus, CCRF-CEM cells incorporate Dex faster and retain it more efficiently than GC-resistant Jurkat cells.

#### 3.2.2. Dex Preferentially Targets Mitochondria

The classical mechanism of Dex action involves the activation of GRs and their nuclear translocation, where they regulate the expression of target genes. DexFluo preferentially accumulated in the mitochondria in T-ALL cells and, to a lesser extent, in the nucleus, as evidenced by co-staining with MtRed. This subcellular DexFluo distribution was observed in both CCRF-CEM and Jurkat cells (Figure 2c–e).

The phenomenon of FRET (fluorescence resonance energy transfer) is a process by which radiationless energy transfer occurs from an excited state fluorophore to a second fluorophore nearby. The range over which the energy transfer can take place is limited to approximately 10 nanometers. Upon co-staining, DexFluo fluorescence is apparently quenched by increasing concentrations of MtRed (see Appendix A), suggesting a donor–acceptor interaction within the mitochondria (Appendix A). Notably, only half of the CCRF-CEM cells were able to accumulate DexFluo (Figure 2a,d,e, green trace). DexFluo accumulation depended on the GRs, as GR antagonist Ru486 strongly reduced DexFluo staining (Figure 2e, blue trace).

#### 3.2.3. T-ALL Cells Can Extrude Dex by Alternative Mechanisms

Cancer cells overexpress transporters such as MDR and P-glycoprotein (P-gp), which promote the efflux of chemotherapeutic drugs. To reveal whether the incapacity to retain Dex by a significant fraction of T-ALL population was due to active Dex extrusion, we applied two well-known inhibitors of MDRs and P-gp, probenecid and sulfinpyrazone (250 μM). Contrary to our expectations, the presence of these compounds in the incubation medium did not increase the accumulation of DexFluo. Moreover, after 30 min, we observed a significant decrease in the percentage of DexFluo-stained CCRF-CEM cells (Figure 2e, red trace). Remarkably, in the presence of MDRs and P-gp inhibitors, we also observed the presence of multiple large (≈3 μm) DexFluo-containing particles, accumulating over time in the extracellular space (Figure 2g,h). In the beginning, DexFluo aggregates were formed in sites attached to the plasma membrane. Starting as puncta, they increased in size before being extruded (Figure 2g). Thus, after the inhibition of active transporters MDRs and P-gp, which extrude Dex from leukemic cells, an alternative mechanism of Dex secretion by large extracellular vesicles was activated.

### 3.3. Dex Induces Metabolic Reprogramming in T-ALL Cells

In T-ALL cells, the proto-oncogene *MYC* regulates at least 15% of the total genome, including genes related to cell metabolism and cell cycle (an extensive review of the importance of *MYC* in T-ALL can be found in [27]). In this regard, rapid time-dependent downregulation of *MYC* is reported as one of the earliest (4–24 h) responses to GC treatment [28]. Dex-mediated *MYC* suppression is accompanied by a reduction in glucose metabolism and subsequent cell death [28,29]. Metabolic plasticity is one of the finest adaptations of cancer cells. It is well known that upon reduced glucose metabolism, compensatory mechanisms are activated to support the cell’s bioenergetics and viability. One of the most important auxiliary strategies in ALL metabolism is the use of amino acids, preferentially glutamine. Indeed, it is widely accepted that cancer cells are “addicted to glutamine consumption” and targeting glutamine metabolism has been proposed as an anticancer strategy [30]. Glutaminolysis is particularly important under glucose deprivation, considering that glutamine metabolism allows the replenishment of the tricarboxylic acid (TCA) cycle via α-ketoglutarate production in the absence of the TCA intermediates produced by glycolysis. Another strategy utilized by ALL is the up-regulation of lipolysis and fatty acid oxidation (FAO) in Dex-treated cells [28,31]. In the present work, we observed Dex targeting mitochondria, accompanied by an initial suppression of cell metabolism and reduced proliferation in sensitive cells (Section 3.1 and Section 3.2). Therefore, we then studied the effect of Dex on the main metabolic pathways in T-ALL cells.

#### 3.3.1. Dex Reduces Glucose Uptake in T-ALL

To estimate the glucose uptake, we used a fluorescent glucose analog, 2-NBDG. Glucose transporters control glucose uptake (e.g., GLUT1 for ALL). To validate our method, T-ALL cells were exposed to increasing concentrations of 2-NBDG (0–200 μM; 30/60 min). Upon incubation, both ALL cell lines exhibited increased 2-NBDG fluorescence in a dose-dependent manner, as evidenced by flow cytometry (Appendix A). The incorporation of 2-NBDG by CCRF-CEM and Jurkat cells was concurrently inhibited by glucose (Appendix A). The uptake of 2-NBDG by CCRF-CEM and Jurkat cells was halved after incubation with 1–10 μM Dex (Figure 3a).

A Dex-induced decrease in the glucose supply should result in a decreased glucose metabolism. In line with this proposal, lactate production was reduced in Dex-treated CCRF-CEM cells (Appendix A).

#### 3.3.2. Dex Reduces Glutaminolysis in T-ALL Cells

To test the effect of Dex on glutaminolysis, we evaluated the production of glutamate, a glutamine metabolism intermediate. Under our experimental conditions, Dex (1 μM) did not potentiate glutamate production, but decreased it in a time-dependent manner in both T-ALL cell lines (Figure 3b and Appendix A).

#### 3.3.3. Dex Promotes Lipolysis in T-ALL

Triglycerides are another intracellular source of energy. They are hydrolyzed by lipolysis, yielding fatty acids and glycerol. Fatty acids are further oxidized into acetyl-CoA, which mitochondria use in the TCA cycle. Thus, glycerol production is an indicator of the activation of this alternative pathway. When exposed to Dex (1 μM), T-ALL cells exhibited a time-dependent increase in glycerol production (Figure 3c), which indicates a reprogramming of cellular metabolism towards using fatty acids as an energy source.

#### 3.3.4. Selective Inhibition of Long-Chain FAO Overcomes GC Resistance in T-ALL

Long-chain FAO is essential to the multistep process of ATP production from fatty acids. FAO in T cells is regulated by the carnitine palmitoyltransferase 1A (CPT1A) system [32]. Selective inhibition of CPT1 by its irreversible inhibitor etomoxir (50 µM) alone is not cytotoxic. However, its combination with increasing Dex concentrations (0.3–10 μM) becomes cytotoxic for CCRF-CEM cells and, importantly, even more for GC-resistant Jurkat cells. Dex alone did not cause a significant change in the number of living T-ALL cells at 24 h (Figure 3d).

#### 3.3.5. Dex Increases Retention of Mitochondrial Indicators in T-ALL

Given the preferential localization of DexFluo in the mitochondria, we tested the effects of Dex on mitochondrial function. TMRE is a fluorescent, permeable organic cation that accumulates in healthy (energized) mitochondria, driven by the large negative transmembrane potential difference, ΔΨm. T-ALL cells colocalize TMRE and Mt Green (Figure 3e), corroborating the specificity of the TMRE staining. Incubation with Dex (1 μM; 24 h) resulted in a time-dependent increase in TMRE fluorescence, which is more pronounced in CCRF-CEM than in Jurkat cells (Figure 3f,g). An increase in TMRE fluorescence can be caused by ΔΨm hyperpolarization and/or an increase in mitochondrial mass/number. Dex did not immediately affect ΔΨm (Appendix A). However, after 24 h of Dex treatment, we observed increased retention of MtGreen (non-dependent on ΔΨm) and an even higher relative increase in MtRed retention (dependent of ΔΨm) in both cell lines (Figure 3f,g). Thus, Dex increased both the mitochondrial content and membrane polarization in leukemic cells.

#### 3.3.6. Dex Modulates Intracellular and Mitochondrial Ca^2+^ Levels in T-ALL

To further explore the effect of Dex on mitochondrial function, we transfected Jurkat cells with CEPIA3mt, a Ca^2+^-sensitive fluorescent protein fused to a signal peptide for mitochondrial delivery [33]. It is well known that a moderate increase in mitochondrial Ca^2+^ ([Ca^2+^]_M_) favors mitochondrial metabolism by promoting the activity of several TCA enzymes [34]. Jurkat cells treated with Dex (1 μM; 24 h) presented higher [Ca^2+^]_M_ levels when compared to untreated cells (Figure 3h). Other ALL cell models display such alterations after Dex treatment, possibly as a compensatory strategy for Dex-induced metabolic disturbances [28]. Intracellular Ca^2+^ not only favors T cell metabolism, but it is also an important second messenger that promotes gene transcription. Transient elevations of cytosolic Ca^2+^ ([Ca^2+^]_C_) have been previously observed in multiple models after Dex administration and were attributed to the store-operated calcium entry (SOCE) to regulate gene expression [35,36,37]. We also recorded rapid changes in [Ca^2+^]_M_ and [Ca^2+^]_C_ in response to Dex (Figure 3i,j). Dex (1 μM) application elicited transient increases in fluorescence related to [Ca^2+^]_C_ and [Ca^2+^]_M_, which were of higher magnitude in Jurkat than in CCRF-CEM cells. Effects of Dex on [Ca^2+^]_C_ and [Ca^2+^]_M_ were not prevented by a selective GR antagonist Ru486, even at a high (10 µM) concentration (Figure 3k).

### 3.4. Oxidative Stress Caused by Dex Contributes to GC Sensitivity in T-ALL Cells

#### 3.4.1. Dex Increases the Production of Reactive Oxygen Species (ROS) by T-ALL Cells

ROS are normal by-products of metabolically active cells, and mitochondria are the primary source of intracellular ROS. Given that in leukemic cells, Dex targets the mitochondria, we evaluated the effect of Dex on ROS production. T-ALL cells under control conditions showed a certain basal level of ROS production, as evidenced by a moderate fluorescence of the ROS-sensitive fluorophore DCF (Figure 4a). Upon Dex treatment, DCF fluorescence increased significantly in CCRF-CEM cells, which remained stable for at least 48 h (Figure 4a,b). Jurkat cells presented a higher basal level of DCF fluorescence and responded with a much smaller increase at 24 h (Figure 4c,d). The Dex-induced increase in ROS production was prevented by Ru486, suggesting that this process is dependent on GR stimulation (Figure 4b,d). As expected, the antioxidant NAC (1–5 mM) efficiently reduced Dex-induced ROS production in a dose-dependent manner (Figure 4e).

#### 3.4.2. Oxidative Stress Caused by Dex Promotes Autophagy in T-ALL Cells

Autophagy is a degradative and protective process regulated by diverse mechanisms, including a challenging metabolic environment, nutrient starvation, or enhanced levels of intracellular ROS [21,38]. To test whether Dex-induced oxidative stress contributes to autophagy induction, we stained the cells with MDC, an acidotropic fluorophore that labels acid endosomes, lysosomes, and late-stage autophagosomes. MDC is commonly used to monitor autophagy because of its preferential accumulation in autophagic vacuoles due to a combination of ion trapping and specific interactions with membrane lipids [39]. In T-ALL cells under basal conditions, MDC fluorescence is dim and diffuse, whereas Dex (10 μM) promotes the accumulation of MDC as multiple discrete highly fluorescent cytosolic puncta (Figure 4f). The antioxidant NAC suppressed the Dex-induced increase in the MDC fluorescence in a dose-dependent manner (Figure 4g). Notably, NAC opposes the cytotoxic effects of Dex in CCRF-CEM cells, whereas Jurkat cells’ viability was neither sensitive to Dex nor to NAC treatment (Figure 4h). Thus, Dex-induced ROS increases upstream of autophagy stimulation and cell death. Our observations align with reports that Dex induces ROS-dependent autophagy [40,41,42].

#### 3.4.3. ROS-Mediated Autophagy Influences GC Sensitivity

To determine if the oxidative status of the Dex-treated cells contributes to the cytotoxic effect of Dex, we evaluated the cell survival in the presence of Dex at different NAC concentrations (0–5 mM). At 72 h, NAC partially limited the cytotoxic effect of Dex in CCRF-CEM cells in a dose-dependent manner (Figure 4h). In Jurkat cells, neither Dex nor NAC affected cell viability.

### 3.5. Dex Promotes Mitochondrial Fragmentation and Mitophagy

#### 3.5.1. TEM Monitoring of Dex-Induced Morphological Changes in Sensitive CCRF-CEM Cells

Although MDC staining is convenient for a quick autophagy assay, results can be misinterpreted because MDC can also stain structures unrelated to autophagy. TEM represents one of the most accurate and informative tools for autophagy monitoring, as it reveals the autophagic structures and their interaction with organelles at a high resolution [21]. Therefore, we applied TEM to monitor structural changes in Dex-treated cells. Under control conditions, CCRF-CEM cells displayed characteristic features of lymphoblasts (Figure 5a, upper panel), that is, round shape with a high nuclear/cytosol ratio, presence of clearly defined cytosolic components such as nuclei (specified by N), lysosomes as dense round structures (indicated by yellow arrows), and rod-like mitochondria (indicated by green arrows) with intact cristae.

Under Dex exposure (1 μM; 24 h), CCRF-CEM cells exhibited dramatic subcellular changes (Figure 5a, lower panel). First, autophagy was corroborated by the accumulation of autophagosomes, evidenced as double-membrane structures with intraluminal content (indicated by the purple asterisk). Mitochondrial morphology was severely affected by Dex. There was a significant reduction in mitochondrial size and an increase in mitochondrial number, which could be a result of fission. Indeed, fission-like processes were clearly observed (red squares). Dex treatment affected the mitochondrial cristae integrity. Lysosomes were less dense than in untreated cells and localized in the proximity of mitochondria or autophagosomes (blue squares).

#### 3.5.2. Dex-Dependent Reduction in Mitochondrial Size and Increase in Mitochondrial Number Is Not Restricted to T-ALL Cells

Fluorescent organelle indicators represent a powerful tool for monitoring their status and subcellular localization. However, evaluation of mitochondrial morphology and distribution by fluorescent microscopy is somewhat complicated in leukemic cells because of a) the large nucleus, which occupies approximately 80% of the cell volume, meaning that changes in cytosolic structures or processes can be masked, and b) the limited number of mitochondria in comparison to other cell types. Therefore, for comparison, we investigated the effect of Dex on mitochondrial size and their distribution in larger adherent spreading cells with a higher mitochondrial content (Figure 5b,c and Appendix A).

For these experiments, we used MCF-7 breast cancer cells and non-oncologic BM stromal cells OP9. Cells were treated with Dex (1 μM, 24 h) and stained with MtRed (200 nM) to visualize mitochondria (Figure 5b). Control cells were efficiently stained and displayed large tubular and hyper-fused mitochondria. Like in T-ALL cells (Figure 5a), Dex treatment caused a significant reduction in mitochondrial size and increased their number, as measured by the number and size of MtRed particles (Figure 5b). Similar effects were observed in MDA-MB-231, OP9, HEK-293 and MCF-7 cells (Appendix A). Therefore, Dex affects the morphology of mitochondria and causes their fission or fragmentation in different cell models.

#### 3.5.3. Autophagic Components Are in Close Contact with Fragmented Mitochondria in Dex-Treated Cells: In Vivo Studies

In TEM images of Dex-treated leukemic cells, we observed the presence of large autophagosomes and double-membrane structures interacting with mitochondria remnants, which could be identified by retaining their appearance cristae (Figure 5a, low panel, third image). Fluorescent confocal microscopy of living cells was used in addition to TEM (fixed cells) to explore the relationship of autophagy with mitochondria fission. To assess whether mitochondrial fragmentation is associated with Dex-induced autophagy, we monitored the interaction of acidic autophagic structures (autophagosomes, lysosomes, endosomes) with mitochondria.

Remarkably, in Dex-treated cells (1 μM, 24 h), autophagic acidophilic markers (e.g., MDC or lysotracker) were not randomly distributed; instead, the location of MDC/lysotracker dots spatially corresponded to the location of fragmented mitochondria (Figure 5c, indicated by white arrows), in agreement with our TEM results in leukemic cells. Similar results have been obtained in several cell lines using MDC and lysotracker staining (Figure 5c, right and Appendix A). In all cases, ring-shaped/spheroid mitochondria (or mitochondrial fragments) were localized in the proximity of lysosomes or autophagosomes in Dex-treated cells, as is expected during mitophagy (discussed in [21]).

#### 3.5.4. Dex Promotes Mitophagy in T-ALL Cells: mtKeima-Based In Vivo Monitoring

A specific strategy to monitor mitophagy in living cells is based on the pH-dependent change in fluorescence of the coral-derived protein Keima, which is resistant to lysosomal degradation and, through genetic engineering, can be targeted to mitochondria (mtKeima) [23]. The mtKeima excitation spectrum peaks at 440 nm at neutral pH and shifts to 586 nm at acidic pH (Figure 5d). The emission spectrum is pH-independent. This allows the discrimination between mtKeima-stained mitochondria in the neutral cytosol and acidic microenvironment of autophagolysosomes (Figure 5d). Mitochondria of CCRF-CEM cells under control conditions mainly localized in the cytosol (Figure 5e). Upon treatment with Dex, mtKeima fluorescence emission at 408 nm excitation diminished, and fluorescence emission at 555 nm excitation increased, indicating the translocation of mitochondria to an acidic microenvironment. Remarkably, the size and distribution of fluorescent organelles also changed from well-defined and large at neutral pH, to small and randomly dispersed at acidic pH. These results are consistent with mitophagy, the fusion of acid lysosomes with damaged, fragmented mitochondria previously engulfed by autophagosomes.

Similar results were obtained when CCCP (carbonyl cyanide *m*-chlorophenyl hydrazone, a mobile proton carrier that collapses ΔΨm) was used to monitor mitochondrial damage. In mtKeima-transfected T-ALL cells, spectrofluorometry was used to analyze Dex-induced mitophagy (Figure 5f) quantitatively. Mitophagy levels were defined as the ratio of maximum fluorescence obtained at 586 nm and 440 nm excitation (586/440). Mitophagy caused by either Dex or CCCP was higher in GC-sensitive CCFR-CEM cells than in GC-resistant Jurkat cells. Dex-induced mitophagy diminished by treatment with the antioxidant agent NAC (5 mM). Our data suggest that Dex-induced oxidative stress, secondary to mitochondrial collapse, promotes autophagy/mitophagy, perhaps as a compensatory mechanism to reduce cellular damage.

#### 3.5.5. Autophagy Inhibition and Mitochondria-Targeted Drugs Sensitize T-ALL Cells to Dex

Autophagy/mitophagy may play a dual role in cancer cell response to cytotoxic drugs: It may represent either (1) a cell defense or (2) a cell death mechanism (reviewed in [43]). To assess the role of autophagy in Dex resistance/sensitivity, we treated T-ALL cells with Dex in the presence of CQ, an autophagy-inhibiting drug (1 µM) [19]. Under these conditions, the percentage of Dex-sensitive (dead) cells increased in CCRF-CEM cells (Figure 5g). Strikingly, in GC-resistant T-ALL cells (Jurkat and MOLT-3), GC resistance was overcome by co-administration of Dex and CQ, resulting in cell death at levels similar to those found in CCRF-CEM cells (Figure 5g, red traces). Thus, autophagy plays a protective role against Dex treatment in T-ALL cells, providing its level does not exceed the lethal threshold like in CCRF-CEM (Figure 4g,h).

An alternative strategy to overcome GC resistance could be severe pharmacological damage to mitochondria, leading to excessive autophagy/mitophagy and subsequent cell death. Remarkably, a synergic cytotoxic effect was observed when mitochondria-targeted drugs, CBD and curcumin, which cause the collapse of ΔΨm and mitochondrial dysfunction [20,44], were applied in combination with Dex (Appendix A).

### 3.6. Dex Reduces the Ability of T-ALL to Migrate and Colonize MSC Niches

#### Dex Inhibits T-ALL Interaction with MSCs and Promotes Cell Death

MSCs are essential components of the bone marrow (BM) microenvironment. In the case of T-ALL cells, interactions between mesenchymal and leukemic cells are believed to limit the toxic effects of chemotherapeutic agents [45].

To assess whether Dex alters the ability of T-ALL to interact with MSCs, we established a 2D-CC system of mesenchymal OP9-GFP cells and leukemic CCRF-CEM cells stained with the Deep Red cell tracker. In the monoculture of OP9 cells, Dex (0–5 µM, 24 h) did not reduce their metabolic rate (Figure 6a), nor their adhesive properties and ability to form a confluent monolayer. Confocal microscopy analysis revealed that, under control conditions, CCRF-CEM cells tend to interact with superficial MSCs (Figure 6b,c). Interestingly, Dex administration dose-dependently limited the interaction between CCRF-CEM and MSCs. Accordingly, in the presence of Dex, leukemic cells were found predominantly in the supernatant rather than attached to MSCs (Figure 6d,e). Moreover, there was a significant increase in dead cells and debris in supernatants in Dex-treated 2D-CCs (Figure 6c,f).

Considering the results obtained with 2D-CC, we inferred that the 2D-CC system might not provide optimal conditions for physical intercellular interaction and protection of T-ALL cells since the thickness of the monolayer barely reaches 50 μm. Therefore, we generated a 3D-CC system (MSC spheroids), approximating leukemic niches in BM (Figure 6g). The creation of larger 3D structures and invasion of leukemic cells to the interior of the MSC spheroid could potentially shield T-ALL against drug exposure. When using 2.5 × 10^4^ OP9-GFP cells, spheroids with a diameter of about 0.5 mm were formed. Under control conditions, CCRF-CEM cells were added 24 h later to the spheroids. Most of them were attracted by the spheroid and developed a reasonably thick layer around it (Figure 6h,i). The thickness of this layer was significantly thinner in cultures treated with Dex (Figure 6h,i).

Confocal microscopy allowed us to trace the exact localization of OP9-GFP and leukemic CCRF-CEM (Deep Red stained) cells, and their mutual arrangement in the inner layers and in the spheroid core, where cellular interactions are stronger (Figure 6j–n). When the spheroids were transferred from the culture plate to the microscope chamber, loosely attached superficial leukemia cells were easily shed (note the decrease in the diameter of the spheroids by comparing Figure 6i with Figure 6k), which contributed to better observation of well-attached leukemic cells. In these experiments, it was found that, despite the efficient recruitment of CCRF-CEM cells into spheroids, only a small fraction form tight physical contact with the MSCs. This fraction is also reduced by Dex (Figure 6j,k). The Deep Red layer diameter (size of the Deep Red circumference at the largest *Z*-axis projection) decreased in Dex-treated spheroids, indicating fewer leukemic cells located in the surface layers. At the same time, analysis of the Deep Red fluorescence intensity distribution profile shows that a lower concentration of Dex (1 µM) led to an increase in fluorescence intensity at deeper inner layers, indicating that under these conditions, leukemic cells moved from the surface to the spheroid core (Figure 6l–n, compare the black and pink profiles). When treated with a high concentration of Dex (10 µM), the Deep Red fluorescence intensity in the spheroid core was less than with 1 µM Dex, which can reflect a decrease in chemokine signaling, mobility or, eventually, an increase in CCRF-CEM cell death in this region.

To verify whether the smaller number of CCRF-CEM cells in the spheroid core was due to the effect of Dex on T-ALL cells or reduced production of chemoattractants by MSCs, we performed a Transwell migration assay, where either CCRF-CEM or OP9 cells were pretreated with Dex (1 μM, 24 h) (Figure 6o). The untreated OP9 recruited leukemic cells to the same extent as CXCL12, a chemokine produced by stromal cells in leukemic niches in BM. Pre-incubation of OP9 cells with Dex did not affect their ability to recruit CCRF-CEM. In contrast, Dex-treated CCRF-CEM cells decreased their capacity to migrate toward OP9 cells or increased the CXCL12 gradient. Taken together, our results demonstrate that Dex limited the ability of leukemic cells to interact with the MSCs and that CC systems in bi- and tridimensional configurations did not protect T-ALL against the effects of Dex treatment.

## 4. Discussion

The main goal of our study was to understand metabolic changes, alteration of mitochondrial function, and induction of autophagy/mitophagy caused by Dex in two T-ALL cell lines to reveal the impact of these changes on GC resistance as a first step in a way to overcome it. The GC-resistant Jurkat cell line and heterogenous cell line CCRF-CEM, which contains clones with different sensitivity to GC, were used as models.

As the results of Figure 1j,l clearly demonstrate, the cytotoxic effects of Dex on the GC-sensitive population of CCRF-CEM cells are mediated by GRs. It has long been known that Jurkat cells have a lower GR expression and a lower induction of GR expression by Dex than the CCRF-CEM cell line [46]. Moreover, selective ectopic expression of some GR isoforms in Jurkat cells sensitized them to Dex treatments [47]. There is accumulated evidence on the differential expression of GR isoforms in different tissues, but their subcellular distributions are less known.

A striking new finding of our study was the rapid and preferential targeting of Dex to mitochondrial compartments in both T-ALL cell lines (Figure 2c–e). In this regard, previous work reported Dex-induced translocation of GRs into mitochondria in GC-sensitive but not in GC-resistant T-lymphoma (multiple) and leukemia (Jurkat) cell lines [15]. There is also evidence that one of the GR isoforms, GRγ, is overexpressed in the mitochondria of GC-resistant cells [48,49]. GRγ accumulates in extra-nuclear localization after Dex treatment and regulates mitochondrial function by increasing the mitochondrial genes and promoting the TCA cycle [50].

Of note, both T-ALL cell lines displayed subpopulations that could or could not accumulate Dex. However, GC-resistant Jurkat cells took up Dex more slowly and extruded it more efficiently than partly sensitive CCRF-CEM (Figure 2a,b). As for Dex cytotoxicity, the Dex uptake by CCRF-CEM was controlled by GRs (Figure 2f).

Dex was shown to decrease glucose metabolism in several cell lines and experimental conditions through classical mechanism, involving the repression of MYC expression, which is directly dependent on GRs, and consequent MYC-dependent decrease in GLUT1 expression and glucose consumption. Considering the high glycolytic phenotype of ALL cells, Buentke and colleagues stated that reduction in glycolysis is responsible for Dex-mediated cell death [29]. Remarkably, Jurkat cells express extremely low levels of nuclear GRα isoform, which is often associated with GC resistance [26]. Accordingly, our data strongly suggest that glucose uptake was slightly affected by Dex in Jurkat and more pronounced in CCRF-CEM cells (Figure 3a), but no cell death was observed at this time (Figure 1). However, the decrease in glucose metabolism per se was insufficient to induce cell death, most likely due to the extreme metabolic plasticity of ALL cells. GC resistance in T-ALL was shown to be associated with a highly proliferative phenotype: increased glycolysis, but also glutaminolysis, FAO, OXPHOS, and mitochondrial biogenesis [9].

In cancer cells, glutaminolysis may replenish the TCA cycle and serves as an additional biosynthetic precursor source [51]. Dex was shown to decrease the entry of glucose and glutamine into the TCA cycle, with an accumulation of glutamine in the cytosol in ALL cells [27]. A possible cause may be the inhibition of glutamate synthesis from glutamine, a necessary step to enter TCA, which is enzymatically regulated by phosphate-dependent glutaminase, whose activity is reduced by Dex [52]. In our experiments, Dex caused a decrease in glutamate production in both cell lines, with a somewhat more significant effect in more susceptible cells (Figure 3b). At the same time, glycerol production (by lipolysis) was increased after Dex treatment to a greater extent in GC-resistant cells (Figure 3c). Prevention of FAO by etomoxir, which inhibits long-chain fatty acid transfer into the mitochondrial matrix, significantly increased the Dex sensitivity of CCRF-CEM cells (Figure 3d). Moreover, etomoxir restored the sensitivity to Dex of GC resistance Jurkat cells, which became even more sensitive to Dex under these conditions than CCRF-CEM (Figure 3d). Therefore, increased fatty acid consumption is essential to T-ALL’s adaptive mechanisms underlying GC resistance. Adipocytes in a co-culture with B-ALL cells are forced by the latter to feed leukemic cells with fatty acids. Consequently, ALL metabolism is switched from glucose to fatty acid oxidation as a source of metabolic energy for OXPHOS, which contributes to chemoresistance [53].

Consequently, inhibition of FAO may be proposed as a therapeutic strategy to address ALL GC resistance. Increased mitochondrial metabolism, OXPHOS, and FAO activation have also been shown to be responsible for chemoresistance in acute myeloid leukemia, AML [54,55]. OXPHOS, namely mitochondrial complex I, was proposed as a rational therapeutic target to improve the efficiency of conventional antileukemic protocols in T-ALL [56].

FAO occurs in the mitochondrial matrix, and its product acetyl-CoA fuels TCA, which produces NADH and FADH_2_, unique substrates for OXPHOS. Our experiments evidenced the hyper-functional status of mitochondria after Dex treatment by increasing [Ca^2+^]_M_ and ΔΨm levels (Figure 3). Moderately high [Ca^2+^]_M_ levels stimulate the activity of mitochondrial dehydrogenases, thus eventually speeding up electron transfer in the respiratory chain [34]. In contrast, enhanced electron-transfer activity results in a high ΔΨm (hyperpolarization), associated with the exponential increase in ROS generation [57]. Dex-induced increase in ROS production was indeed observed (Figure 4e,f). The level of ROS production remained high in more GC-sensitive CCRF-CEM. The basal ROS production in Jurkat was twofold higher than in CCRF-CEM and showed a lesser increase in response to Dex (Figure 4a–d). It appears that CCRF-CEM is more sensitive to ROS and its increase. Therefore, when CEM-CCRF cells were supplied with externally added antioxidants, this restored their viability and proliferative potential under Dex treatment (Figure 4h). Therefore, the inability to handle the Dex-induced ROS challenge is an important feature contributing to GC sensitivity.

Increased ROS production can cause cell damage and trigger autophagy to recycle damaged organelles [58,59,60]. Dex induced autophagy in both leukemic cell lines (Figure 4g). In CCRF-CEM cells, the antioxidant NAC effectively prevented autophagy and cell death, indicating oxidative stress to be an essential contributor to lethal autophagy (Figure 4g,h). In contrast, in GC-resistant Jurkat cells, NAC had little effect on Dex-induced autophagy and survival/proliferation (Figure 4g,h). Moreover, the inhibition of autophagy by CQ made Jurkat cells even more sensitive to Dex than CCRF-CEM. Thus, Dex-induced autophagy is associated with a pro-death scenario in Dex-treated sensitive T-ALL cells and a pro-survival one in Dex-resistant ones. Remarkably, Dex-induced autophagy also preceded apoptotic cell death in GC-sensitive B-ALL cell lines [40]. The inhibition of autophagy by CQ significantly sensitized T-ALL cells to the Dex treatment (Figure 5g), similar to an earlier reported study carried out on the lymphoma cell lines Raji (Burrkitt) and U-937 (histiocytic) [61]. Interestingly, not only inhibition, but also induction of autophagy by tamoxifen [19] or BH3 mimetic obatoclax [62] reversed GC resistance in ALL cells. Altogether, these data evidence that an “optimal” autophagy level seems to be required to resist drug cytotoxicity, and both inhibition and stimulation can help to overcome drug resistance. However, our knowledge regarding the role of autophagy in ALL is still insufficient and further research is required. Because autophagy may contribute to both death and survival in acute leukemias, the use of autophagy modulators needs to be carefully evaluated [63].

TEM revealed that mitochondria are the organelles most affected by Dex treatment. Mitochondria were severely damaged and fragmented (Figure 5a). Damaged mitochondria were then subjected to mitophagy, which was confirmed by both TEM and mitophagy-specific methods based on the use of mtKeima in living cells (Figure 5a–f). Dex-induced mitochondrial fission appears to be a general effect, also observed for other cell types, as shown here (Figure 5b) and elsewhere. Studies on muscle cells revealed that Dex upregulated one of the genes responsible for mitochondrial fission. In another study, Dex upregulated the expression of the Parkin protein, whose activity degrades the mitochondrial fusion proteins [64,65].

It is well known that if the damage is too extensive, an increased level of autophagy causes cell death. In this case, we hypothesize that the mitochondria-targeted drugs, which can increase mitochondrial damage, decrease the threshold for other cytotoxic drugs affecting mitochondria. Indeed, two mitochondria-targeted drugs (mitocans), CBD and curcumin [43], exhibited synergism with Dex (Appendix A).

There is abundant evidence that the leukemic niche in BM, via different forms of interaction between leukemic and MSCs, such as chemical signaling, metabolic exchange, secretion and acquisition of extracellular vesicles, direct adhesion, or contacts via nanotubes, generally exert a chemoprotective effect on AML and B-ALL blasts [66,67,68,69]. In particular, chemoresistance can be associated with the transfer of active mitochondria from BM MSCs to AML [70,71,72]. B-ALL cells either employ a similar mechanism [73] or export damaged mitochondria and autophagosomes from ALL blasts to BM MSC [74]. GC-Resistant Jurkat T-ALL cells transfer mitochondria to BM MSCs, which underlies their chemoresistance [75]. Here, we showed the co-culture of GC-sensitive CCRF-CEM with MSCs in 2D/3D-CC systems instead of protection-sensitized T-ALL cells to Dex. Dex reduced the ability of T-ALL cells to migrate and colonize MSC niches and promoted cell death (Figure 6). Of note, 2D-CC of CCRF-CEM with MSCs per se induced apoptosis in T-ALL blasts [76], and the same is true for another GC-sensitive T-ALL line, MOLT-4 [77,78,79].

## 5. Conclusions

A summary of Dex-induced changes in T-ALL cells and differences between GC-resistant and GC-sensitive cell responses are presented in Figure 7. We have demonstrated that Dex targets mitochondria in T-ALL. Metabolic reprogramming towards lipolysis and FAO was observed in treated cells, to overcome decreased glycolysis and glutaminolysis. Dex-induced enhanced mitochondrial metabolism caused oxidative stress, mitochondrial damage, and mitophagy, which contributed to cell death in GC-sensitive cells. It appears that the degree of these injuries was lower in GC-resistant cells, where autophagy/mitophagy was associated with a pro-survival scenario. In this case, FAO and mitophagy prevention and/or the use of mitocans may represent convenient strategies to increase GC cytotoxicity and sensitize GC-resistant leukemic cells to GC-based chemotherapy. The role of metabolic reprogramming and autophagy/mitophagy, which are important for leukemic cell plasticity in response to drug treatment, are worth being studied in a wider context, including more phenotypes of leukemia and cancers in general, to reveal the underlying mechanisms in detail and to evaluate the translational capacity.

## Figures and Tables

**Figure 1 cancers-15-00445-f001:**
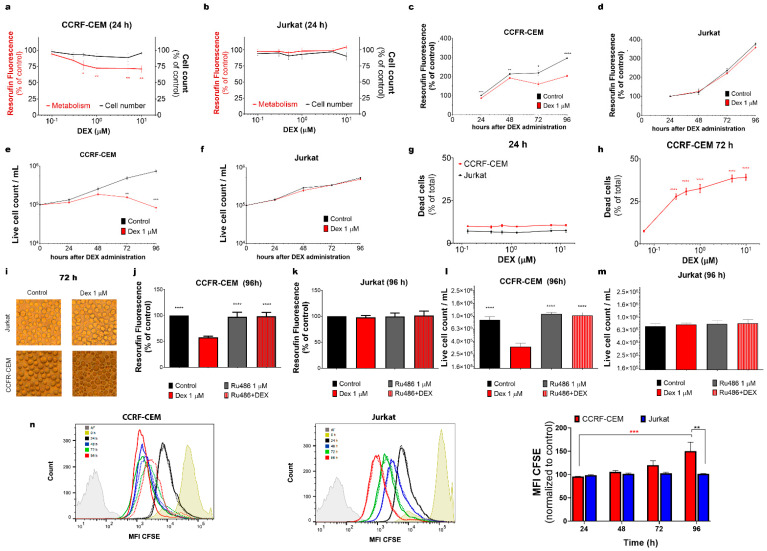
Validation of T-ALL models for GC resistance. (**a**,**b**) The effect of Dex (0–10 μM) on cell metabolism (red traces) and cell count (black traces) was estimated at 24 h in CCRF-CEM (**a**) and Jurkat cells (**b**). One-way ANOVA with Dunnet’s multiple comparison test was employed. (**c**,**d**) The effect of Dex (1 μM) on cell metabolism as a function of time (0–96 h) in CCRF-CEM (**c**) and Jurkat cells (**d**). (**e**,**f**) The effect of Dex (1 μM) on live cell count was monitored (0–96 h) in CCRF-CEM (**e**) and Jurkat cells (**f**) using the trypan blue exclusion test. In (**c**–**f**), multiple *t*-tests were employed for each time point to compare control vs. treated groups. (**g**,**h**) Cell death of Dex (0–10 μM)-treated cells was estimated using flow cytometry at 24 h (**g**) and 72 h (**h**). Total cell death includes apoptotic and necrotic cells, as evidenced by AnnexinV- Alexa488 and/and PI inclusion, respectively. One-way ANOVA with Dunnet’s multiple comparison test was employed. (**i**) Representative light microscopy micrographs (10×) of Jurkat and CCRF-CEM cells exposed to Dex (1 μM, 72 h, right) in comparison to control (72 h, left). (**j**,**k**) Effects of Dex, Ru486, or their combination (1 μM, 96 h) on CCRF-CEM (**j**) and Jurkat (**k**) metabolism. (**l**,**m**) Effects of Dex (1 μM), Ru486 (1 μM) or their combination on live cell count (96 h), in CCRF-CEM (**l**) and Jurkat (**m**). One-way ANOVA with Dunnet’s multiple comparison test was employed. (**n**) Cell proliferation estimated using CFSE-based flow cytometry in CCRF-CEM and Jurkat cells. Representative histograms for CCFR-CEM (left) and Jurkat (right) demonstrate the progress of cell proliferation as a leftward shift of the mean fluorescence intensity (MFI), reflecting cell divisions by dye dilution. Solid and dashed lines are for Dex-treated and control cells, respectively. Bar-chart at the right displays the ratio Dex/control MFI at each time. Two-way ANOVA with Sidak’s multiple comparison test was employed to compare relative MFI values for Jurkat and CCRF-CEM at different times against MFI at 24 h. In (**a**–**d**,**g**,**h**,**j**–**n**), raw data were normalized to the control group. Wherever applicable, data are mean ± S.E. for 3 to 7 independent experiments. * *p* < 0.05; ** *p* < 0.01; *** *p* < 0.001; **** *p* < 0.0001.

**Figure 2 cancers-15-00445-f002:**
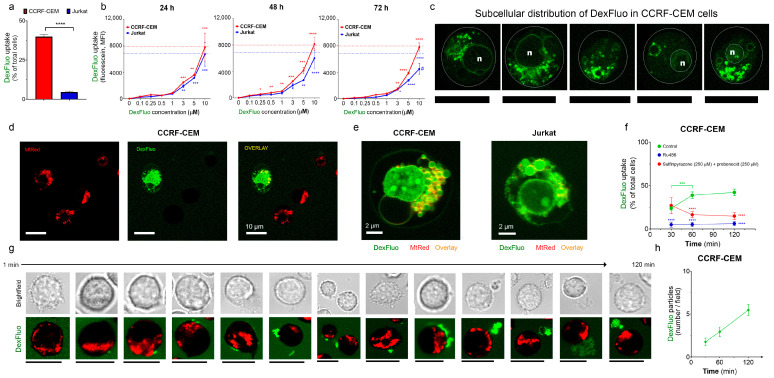
DexFluo uptake and subcellular distribution by T-ALL cells. (**a**) Quantitative analysis of the incorporation of DexFluo (5 μM, 1 h) using confocal microscopy (*n* = 4, total of 975 (CCRF-CEM) and 1252 (Jurkat) cells). (**b**) DexFluo accumulation as a function of its concentration in the medium was estimated using flow cytometry at different incubation times (0–72 h). One-way ANOVA with Dunnet’s multiple comparison test was employed. *—comparison vs. control group, #—comparison 72 h vs. 24 h. (**c**) Representative images (40×) of intracellular DexFluo (1 μM; 24 h) distribution in CCRF-CEM cells. Nuclei: n. Scale bar: 10 μm. (**d**) Representative fluorescent micrographs of CCRF-CEM cells exposed to for 2 h to DexFluo (5 μM, green staining) and the mitochondrial probe MtRed (80 nM, red fluorescence). Objective: 40×. Scale bar: 10 μm. (**e**) Representative high magnification (100×) fluorescence overlay images of CCRF-CEM (left) and Jurkat (right) cells after incorporation of DexFluo (5 μM) and MtRed (80 nM) for 2 h. Scale bar: 2 μm. (**f**) Quantitative analysis of DexFluo uptake using CCRF-CEM by confocal microscopy. Green trace: DexFluo (5 μM), *n* = 4, a total of 1105 cells (30 min: 386; 1 h: 374; 2 h: 345 cells); blue trace: DexFluo (5 μM) in the presence of the GR antagonist Ru486 (5 μM; 20 min pre-incubation), *n* = 4, a total of 1898 cells (30 min: 634; 1 h: 657; 2 h: 610 cells); red trace: DexFluo (5 μM) after 20 min pre-incubation in the presence of sulfinpyrazone (250 μM) and probenecid (250 μM), *n* = 4, a total of 1824 cells (30 min: 384; 1 h: 748; 2 h: 692 cells). Two-way ANOVA with Dunnet’s multiple comparison test was employed. Statistical significance against the control group is depicted. (**g**) Examples of the time course of DexFluo (5 μM; 0–120 min) extrusion by CCRF-CEM cells. DexFluo particles are green and mitochondria stained with MtRed (80 nM) are red. Scale bar: 10 μm. (**h**) Quantification of the number of green extracellular particles extruded from DexFluo-treated CCRF-CEM cells over 2 h of observation using confocal microscopy. In (**a**–**c**,**f**,**h**), data are mean ± S.E. for at least 3 independent experiments. ** *p* < 0.01; *** *p* < 0.001; **** *p* < 0.0001.

**Figure 3 cancers-15-00445-f003:**
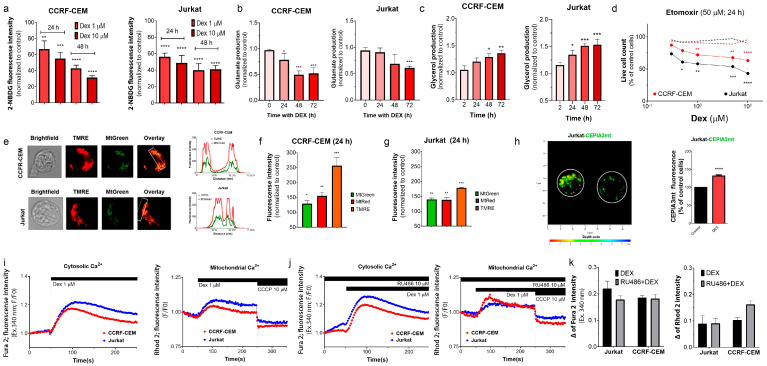
Dex-induced metabolic reprogramming of T-ALL cells. (**a**) Fluorometric analysis of 2-NDBG uptake in T-ALL cells exposed to Dex (1 μM, 24–48 h). An independent *t*-test was performed for each condition. (**b**) Glutamate production was measured through luminometry of T-ALL cells exposed to Dex (1 μM, 24–72 h). A one-way ANOVA test was employed. (**c**) Glycerol production was measured through luminometry of T-ALL cells exposed to Dex (1 μM, 24–72 h). A one-way ANOVA test was employed. (**d**) Effect of various concentrations of Dex alone or combined with etomoxir (50 µM) on live cell count in CCFR-CEM and Jurkat cells after 24 h of incubation with drugs. For Dex alone, data are redrawn from Figure 1a,b (dashed lines). For each timepoint, multiple *t*-tests were employed to compare control with treated groups. (**e**) Representative micrographs of T-ALL cells co-stained with TMRE and Mt Green are shown with phase contrast, TMRE, and MtGreen fluorescence and their overlay, respectively. The fluorescent profiles shown at the right were extracted from the selected sections (white rectangles) of stained cells. (**f**,**g**) Retention of different mitochondrial indicators by CCFR-CEM (**e**) and Jurkat (**f**) cells exposed to Dex (1 μM, 24). (**h**) Representative micrograph of Jurkat cells, transfected with the mitochondrial Ca^2+^-sensitive protein CEPIA3mt (left). Effect of Dex (1 μM, 24 h) on the fluorescence of CEPIA3mt in Jurkat cells (right). For (**f**–**h**), a *t*-test was performed. One hundred cells were analyzed in each experiment for each condition. (**i**) Changes of cytosolic free (left) and mitochondrial (right) Ca^2+^ levels in CCFR-CEM and Jurkat cells in response to Dex (1 μM). (**j**) The same as in (**h**), but in the presence of the GR antagonist Ru486 (20 min preincubation). (**k**) Ru486 did not affect the intracellular Ca^2+^ response to Dex. Δ Ca^2^ in the ordinate is the difference between peak response and basal Ca^2+^ level. An independent *t*-test was performed for each condition. In (**a**–**d**,**f**–**k**), raw data are mean ± S.E. for at least 3 independent experiments. * *p* < 0.05; ** *p* < 0.01; *** *p* < 0.001; **** *p* < 0.0001.

**Figure 4 cancers-15-00445-f004:**
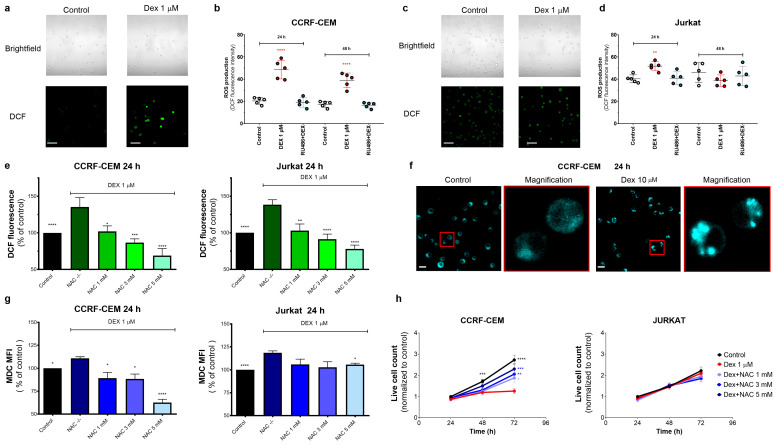
Effects of Dex on ROS production and autophagy. (**a**) Representative confocal micrographs of CCRF-CEM cells stained with the ROS-sensitive fluorophore DCF (20 μM) after treatment with Dex alone or combined with Ru486 (1 μM, 24 h). Scale bar: 40 μm. (**b**) Dex-induced ROS production in CCRF-CEM cells is prevented by Ru486 (1 μM, 24/48 h). One-way ANOVA with Dunnet’s multiple comparison test was executed. (**c**) As in (**a**), but with Jurkat cells. (**d**) As in (**b**), but with Jurkat cells. (**e**) Effect of NAC (1–5 mM) on the Dex (1 μM, 24 h) induced ROS production in CCRF-CEM (left) or Jurkat (right) cells. One-way ANOVA with Dunnet’s multiple comparison test was performed. (**f**) Representative confocal micrographs of CCRF-CEM cells, stained with MDC (60 μM) under control conditions (left) or treated with Dex (right). Scale bar: 10 μm. (**g**) Dex (1 μM, 24 h) induced MDC retention in the presence of NAC (1–5 mM) in CCRF-CEM (left) or Jurkat cells (right). One-way ANOVA with Dunnet’s multiple comparison test was applied. (**h**) Effect of NAC on Dex-induced cytotoxic effect. A trypan blue exclusion test was used to identify the living cells. One-way ANOVA with Dunnet’s multiple comparison test was applied. For (**e**,**g**,**h**), data are mean ± S.E. of at least 3 independent experiments. * *p* < 0.05; ** *p* < 0.01; *** *p* < 0.001; **** *p* < 0.0001.

**Figure 5 cancers-15-00445-f005:**
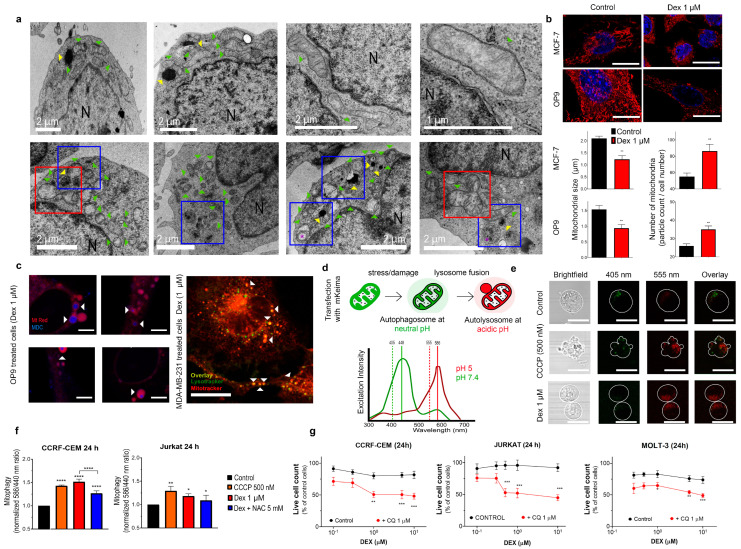
Dex causes structural changes in mitochondria and induces mitophagy. (**a**) High-resolution electron micrographs of CCRF-CEM cells under control (upper panel) and after Dex treatment (lower panel, 1 μM, 24 h). N: nuclei. Green arrows: mitochondria, yellow arrows: lysosomes, purple asterisks: double-membrane autophagosomes., red squares: mitochondria undergoing the fission process, blue squares: autophagic components close to lysosomes. Scale bars: 2 μm. (**b**) Representative confocal images of MCF-7 and OP9 cells co-stained with MtRed and Hoechst, either untreated or treated with Dex (upper panel). Scale bars: 10 μm. Quantitative analysis of MtRed-positive particles’ size and number (lower panel). Independent *t*-tests were performed. (**c**) Confocal images of OP9 (left) or MDA-MB-231 cells (right) co-stained with MtRed (red), MDC (blue), and Lysotracker (green). OP9 micrographs (left) show different zones in Z-stacked layers of a representative cell. White arrows indicate the proximity of mitochondria to acidic organelles. Scale bars: 2 μm (OP9) and 10 μm (MDA-MB-231). (**d**) Experimental strategy for the mitophagy detection with mtKeima (upper panel). The excitation spectrum of mtKeima at neutral (green) and acidic (red) pH (lower panel). Dotted lines: wavelengths corresponding to lasers used for excitation in confocal microscopy. (**e**) Representative confocal microscopy micrographs of CCRF-CEM cells transfected with mtKeima, untreated (control), and treated either with Dex (24 h) or CCCP (positive control). Scale bars: 10 μm. (**f**) Quantitative analysis of mitophagy through spectrofluorometry. The level of mitophagy was calculated as the ratio between the values of the maximum fluorescence obtained upon excitation at 586 nm and 440 nm (ratio 586/440). One-way ANOVA with Dunnet’s multiple comparison test was run. (**g**) Cell viability by trypan blue exclusion assay in T-ALL cells exposed to Dex alone (0–10 μM) and in combination with CQ (1 μM) for 24 h. Two-way ANOVA and Sidak’s multiple comparison tests were employed. In (**b**,**f**,**g**), data are mean ± S.E. for at least 4 independent experiments. * *p* < 0.05; ** *p* < 0.01; *** *p* < 0.001; **** *p* < 0.0001.

**Figure 6 cancers-15-00445-f006:**
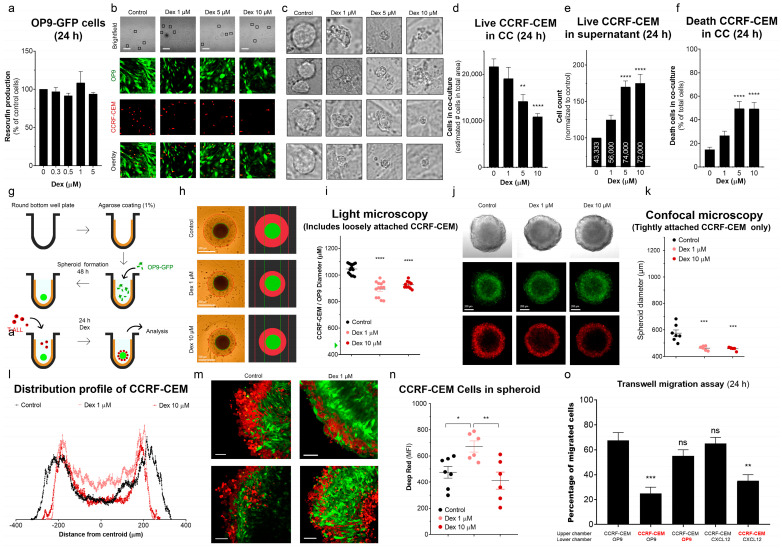
Effects of Dex on T-ALL in (CCRF-CEM) MSCs (OP9) co-cultures. (**a**) Effect of Dex (0–5 μM, 24 h) on the metabolism of the OP9-GFP cells. Data are mean ± S.E. of at least 3 independent experiments. Raw data were normalized to control. An independent *t*-test was performed. (**b**) Representative micrographs of 2D-CC, either untreated or exposed to Dex (24 h). Green: OP9-GFP cells, red: CCRF-CEM cells stained with Deep Red cell tracker. Scale bars: 50 μm. (**c**) Representative morphology of CCRF-CEM cells on the surface of 2D-CC (from b, brightfield, black squares). (**d**) Quantification of the living CCRF-CEM cells, interacting with the MSCs monolayer, in untreated and Dex-treated (24 h) 2D-CC. (**e**) Quantification of the living CCRF-CEM cells in the supernatant of 2D-CC, either untreated or after exposure for 24 h to Dex. (**f**) Quantification of the dead CCRF-CEM cells in the supernatant of 2D-CC, untreated or after Dex exposure (24 h). In (d-f), an independent *t*-test was performed. (**g**) Experimental strategy to obtain 3D-CC. (**h**) Representative light microscopy images of intact 3D-CC, untreated or Dex-treated (1 and 10 µM, 24 h). Scale bars: 200 μm. (**i**) Quantitation of 3D-CC spheroids’ diameter, untreated or after Dex exposure (1 and 10 µM, 24 h), light microscopy. (**j**) Representative confocal microscopy micrographs of intact 3D-CC spheroids, untreated and exposed to Dex (1 and 10 µM, 24 h). Green: OP9-GFP cells, red: CCRF-CEM cells stained with Deep Red. Scale bars: 200 μm. (**k**) Quantification of 3D-CC spheroids’ diameter, untreated or exposed to Dex (1 and 10 µM, 24 h). Confocal microscopy. (**l**) Distribution profiles of Deep Red fluorescence (corresponding to CCRF-CEM) intensity across 3D-CC spheroid. The histogram is the average of at least 6 independent experiments. (**m**) Representative high-resolution (40×) micrographs of independent 3D-CC spheroids. Green: OP9-GFP cells, red: CCRF-CEM cells stained with Deep Red. Scale bars: 50 µm. (**n**) Quantitative analysis of the mean fluorescence intensity (MFI) of the CCFR-CEM cells retained in the 3D-CC spheroid. One-way ANOVA with multiple comparison tests was used. (**o**) Effect of Dex (1 μM) treatment on the migration of CCRF-CEM cells from the upper to the lower chamber. The experimental strategy is specified under the graph: CCRF-CEM or OP9 cells were either pretreated with Dex (1 μM, 24 h, red font color) or untreated (black font). One-way ANOVA with multiple comparison tests was performed. In (**a**,**d**–**f**,**o**), data are mean ± S.E. for at least 4 independent experiments. * *p* < 0.05; ** *p* < 0.01; *** *p* < 0.001; **** *p* < 0.0001.

**Figure 7 cancers-15-00445-f007:**
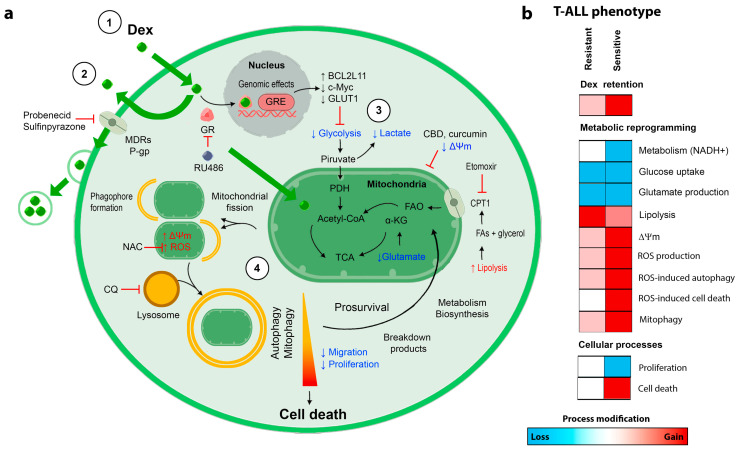
(**a**) Summary of Dex-induced changes in T-ALL cells. (**1**) Dex was rapidly taken up and showed better retention in GC-sensitive CCRF-CEM cells. In addition to well-known canonical mechanisms, mediated by GR, nuclear translocation, and changes in gene expression, Dex preferentially targeted mitochondria in T-ALL. (**2**) T-ALL cells actively extruded Dex. Inhibition of MDRs and P-gp-mediated active extrusion mechanisms by probenecid and sulfinpyrazone induced an alternative pathway for Dex extrusion via the release of Dex-loaded large extracellular vesicles. (**3**) Dex decreased glucose uptake, glycolysis, and glutaminolysis. As a compensatory mechanism, lipid metabolism was increased; lipolysis was up-regulated and T-ALL cells became sensitive to inhibitors of CPT1 (etomoxir), which imports FAs into mitochondria. Dex acted synergistically with compounds, suppressing the mitochondrial function (CBD or curcumin, causing mitochondrial Ca^2+^ overload and collapsing the ΔΨm). (**4**) Changes in mitochondrial metabolism, induced by Dex, favored mitochondrial fission. Mitochondria in Dex-treated T-ALL cells exhibited an increased energization and elevated ROS production. Increased ROS level stimulated autophagy/mitophagy. The degree of autophagy/mitophagy induction determined the cell fate. In CCRF-CEM cells, Dex-induced autophagy suppressed the proliferation, which could be reverted by ROS scavenging with NAC. Excessive autophagy could eventually lead to cell death. A moderate increase in autophagy in Jurkat cells favored a pro-survival scenario, likely via recycling of autophagy products and their usage for biosynthesis and energy production. Lastly, the stoppage of autophagy by CQ caused a dramatic increase in Dex cytotoxicity in both T-ALL cell lines. (**b**) Differences in the responses of GC-resistant and GC-sensitive cell lines to Dex. Color coding: blue means a decrease in the process rate or parameter value, white means no significant change, and red means an upregulation. Abbreviations: GR: glucocorticoid receptors; P-gp: p-glycoprotein; MDR: multidrug resistance proteins; GRE: glucocorticoid response elements; GLUT1: glucose transporter 1; PDH: pyruvate dehydrogenase; FAs: free fatty acids; CPT1: carnitine palmitoyltransferase 1; FAO: fatty acid oxidation; α-KG: alpha ketoglutarate; TCA: tricarboxylic acid cycle; ΔΨm: mitochondrial membrane potential; CQ: chloroquine; ROS: reactive oxygen species; CBD: cannabidiol; NAC: N-acetyl-cysteine.

## Data Availability

The data presented in this study are available within the article and Appendix A.

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
