# Peer review of "Dexamethasone-Induced Fatty Acid Oxidation and Autophagy/Mitophagy Are Essential for T-ALL Glucocorticoid Resistance"

_cancers, 2023, doi:10.3390/cancers15020445_

Round 1

Reviewer 1 Report (Previous Reviewer 1)

The authors present a revised version of their manuscript, responding to most of the concerns raised. Unfortunately, there has been some misunderstanding on several points raised. It remains an opinion of the reviewer that to strengthen the implications of their findings (mostly limited to two extensively grown T-ALL cell lines) other models which may more faithfully represent primary leukemia cells such as patient derived T-ALL cells or xenografts (which can be obtained and successfully cultured in vitro for short periods of time) could have been attempted. It is taken that this is a difficult task, however especially when novel therapeutic combinations are proposed, the use of adequate models should be pursuit.

Given these limitations, the work is well presented and executed.

Author Response

We are grateful to this reviewer for taking the time to review our manuscript and for the comments. The MS was previously revised to address some of the important issues raised. We agree that the results of the present study should be extended in future studies, to include more cell lines, primary leukemic samples, and xenograft models. It is worth noting here that there are some local regulations for obtaining primary samples for the research. We are registered at the Mexican Leukemia Research Database (PRONACES: Programas Nacionales Estrategicas – Health - Leukemia) with our requirements for primary samples which include type of leukemia (T/B-ALL), source (bone marrow/peripheral blood), sample volume, and the type of study we are doing (genetic/functional analyses). Authorized hospitals and clinics provide appropriate samples to research teams, when available. But there are no hospitals/clinics in our city, where leukemic patients are treated (the nearest available ones are in Guadalajara, 300 km away). For our purposes (further cultivation and functional analyses), samples in very good condition (high percentage of live cells), especially, T-ALL primary samples are very rare and in the last three years we have not received a single one. Regretfully, our center does not count with proper installations (vivarium) for the research on xenograft models. So, we were signed at the waiting list of research labs with the national Xenograft Research Lab, located in Mexico City. Hopefully, these links will gain shape, so that fundamental results can be translated to clinics in the future. In response to reviewers’ comments, we have added the final phrase to the conclusions (lines 1033-1036).

Reviewer 2 Report (Previous Reviewer 2)

I appreciate the changes introduced in the article and encourage the authors to further deepen the study of mitophagy in leukemia by extending the study and perform additional screening of  B and T cell line for the observed effects of Dexa.

Author Response

We are grateful to this reviewer for taking the time to review our manuscript and for the comments. The MS was previously revised to address some of the important issues raised. We agree with this reviewer that, bearing in mind its potential importance and novelty, the role of mitophagy in Dex resistance should be extended in future studies, to include more T- and B-ALL cell lines. In response to reviewers’ comments, we have added the final phrase to the conclusions (lines 1033-1036).

This manuscript is a resubmission of an earlier submission. The following is a list of the peer review reports and author responses from that submission.

Round 1

Reviewer 1 Report

In this manuscript, Olivas-Aguirre et al. compare the uptake of dexamethasone (Dex), its intracellular localization and retention together with its effects on some mitochondrial functions in a glucocorticoid sensitive cell line T-ALL cell line (CCRF-CEM) and a glucocorticoid resistant cell line T-ALL cell line (Jurkat). The authors describe (following Dex exposure): preferential targeting of Dex to the mitochondria (both cell lines), metabolic reprogramming such as decreased glucose uptake and glutaminolysis (both cell lines), increased lipolysis (both cell lines), increased oxidative stress (more in sensitive cell line). The authors thus propose autophagy inhibition and general mitochondria-targeting drugs to sensitize T-ALL cells to Dex.

Although the authors present a large amount of data, a major limitation of the study is that it is based on only two T-ALL cell lines, one of which (CCRF-CEM) is known to represent a mixed population of sensitive and resistant clones (Medh R.D et al. 2003). In some studies, CCRF-CEM cells (unselected) are considered rather resistant (if no specific clone is mentioned), although a great variability in Dex IC50 has been reported for this cell line (Beesley, AH et al 2006; Martin-Aragon et al. 2000). For example, in the mentioned paper by Beesley et al. 2006, the IC50 of the CCRF-CEM cell line and Jurkat cells was equal (500 mM). Although the CCRF-CEM cell line used by the authors seems more sensitive than that reported by these authors, it may not represent a highly Dex sensitive cells line as minimal loss of viability is detected after 48h incubation with Dex at 1mM. The authors may consider using T-ALL cell lines such as DND41, HSB2 cells or CEM-C7 cells, known to be Dex sensitive.

The authors state that their goal was to understand metabolic changes, alterations in mitochondrial function and induction of autophagy in sensitive and resistant T-ALL cells. However, notwithstanding the limited number of cell lines analyzed (n=1 for each group) metabolic assays are limited to only colorimetric/fluorometric/luminesce based assays, no steady state or metabolic flux assays are presented which would have been more complete. It would have been very much appreciated an attempt at validating their main findings in primary T-ALL/xenograft samples or additional T-ALL cell lines or results corroborated by additional methods: for example, autophagy analysis could be completed with other methods such documentation of LC3-I to LC3-II conversion and/or p62 degradation.

Another issue lies in the novelty of the findings and the lack of mechanistic insights. Most of the conclusions are not novel but reiterate what is known from the literature, for example targeting autophagy has already been demonstrated to alter glucocorticoid sensitivity (Jiang L et al. 2015, Loane E et al.2009, Bonapace L. 2010, Torres-Lopez L et al. 2019).

Reviewer 2 Report

Olivas-Aguirre et al analyze Dex uptake, intracellular localization, and retention by T-ALL cells, its effect on mitochondrial morphology and function, the metabolic switch, and autophagy/mitophagy induction to achieve a better understanding of how these changes support GC- resistant vs. GC-sensitive phenotype and to find the way to revert the GC-resistance in T-ALL

Excellent manuscript, thoughtfully organized.Great experimental design in material&methods. No major comments.

Minor Comments:

 Materials&Methods:

Line 235, cell-free RMPI simples should be corrected “cell-free RPMI samples”

Line 365 The samples were exited should be corrected excited

In the materials, the lines used are indicated: Human T-ALL cell lines, Jurkat  (ATCC®TIB™, Clone E6-1, male, 14 years), MOLT-3 (ATCC®CRL-1552™, male, 19 years), and CCFR-CEM (ATCC®CCL-119™, female, 4 years)..

And regarding the MOLT-4 line. Why this line is not mentioned in materials but appears in the Figure 5, pag 18. Is it a mistake?

In figure footnotes, some phrases are repeated too many times, such as “Data are mean ± S.E. of at least 3 independent experiments. Raw data were normalized to the control group” which makes reading difficult. They could be unified by indicating, for example Figure, 3, a,b,c: Data are mean ± S.E. of at least 3 independent experiments. Raw data were normalized to the control group.

Why is the MOLT-3 line only described in the Dex and CQ coadministration experiment? Could it be used in other trials? I believe that extending the study to more sensitive and resistant cell lines would enrich the work and give more robustness to the conclusions.